# Multi-Objective Meta Learning

**Feiyang Ye**[1,2,*], **Baijiong Lin**[1,*], **Zhixiong Yue**[1,2], **Pengxin Guo**[1], **Qiao Xiao**[3], and **Yu Zhang**[1,4,†]

[1] Department of Computer Science and Engineering, Southern University of Science and Technology
[2] University of Technology Sydney
[3] Eindhoven University of Technology
[4] Peng Cheng Laboratory
{12060007,linbj,yuezx,12032913}@mail.sustech.edu.cn
{qiaoxiao7282,yu.zhang.ust}@gmail.com

## Abstract

Meta learning with multiple objectives has been attracted much attention recently since many applications need to consider multiple factors when designing learning models. Existing gradient-based works on meta learning with multiple objectives mainly combine multiple objectives into a single objective in a weighted sum manner. This simple strategy usually works but it requires to tune the weights associated with all the objectives, which could be time consuming. Different from those works, in this paper, we propose a gradient-based Multi-Objective Meta Learning (MOML) framework without manually tuning weights. Specifically, MOML formulates the objective function of meta learning with multiple objectives as a Multi-Objective Bi-Level optimization Problem (MOBLP) where the upper-level subproblem is to solve several possibly conflicting objectives for the meta learner. To solve the MOBLP, we devise the first gradient-based optimization algorithm by alternatively solving the lower-level and upper-level subproblems via the gradient descent method and the gradient-based multi-objective optimization method, respectively. Theoretically, we prove the convergence properties of the proposed gradient-based optimization algorithm. Empirically, we show the effectiveness of the proposed MOML framework in several meta learning problems, including few-shot learning, domain adaptation, multi-task learning, and neural architecture search. The source code of MOML is available at `https://github.com/Baijiong-Lin/MOML`.

## 1 Introduction

In the past few years, deep learning has achieved great success in various fields [43] because it can effectively and efficiently process massive and high-dimensional data. However, training a deep learning model from scratch often requires a large amount of labeled data to learn a large number of model parameters and needs to choose hyperparameters by hand.

As a way to address those problems by enabling models to learn how to learn, meta learning has attracted considerable attention recently [19, 20]. Meta learning gains knowledge from multiple meta training tasks so that the knowledge can be reused in new tasks or new environments rapidly with a few training examples. Taken broadly, objective functions of meta learning models are usually formulated as a bi-level optimization problem where the lower-level subproblem represents the adaptation to a given task with learned meta parameters and the upper-level subproblem tries to optimize these meta parameters via a meta objective [19]. Hence, from this view, meta learning has a wide range of applications such as hyperparameter optimization [13], Neural Architecture Search (NAS) [29], and Reinforcement Learning (RL) [68].

---

[*]Equal contribution.
[†]Corresponding author.

35th Conference on Neural Information Processing Systems (NeurIPS 2021).

In many studies on conventional meta learning methods and applications, there is only a single meta objective in the upper-level subproblem. For example, the Model-Agnostic Meta-Learning (MAML) method [12] only measures the performance on a validation dataset in the upper-level subproblem to evaluate the learned initialization of parameters. DARTS [29], a differentiable method for NAS, evaluates the performance of the searched architecture on the validation dataset. However, in real-world applications, there are usually more than one objective to be considered. For example, for MAML, we may need to consider not only the performance but also the robustness which can help adapt to new tasks with the learned initialization. Similarly, the network size and performance should be balanced in NAS, especially when the searched architecture will be deployed to devices with limited resources such as mobile phones. In those applications, we can see that there is a need to balance multiple possibly conflicting objectives in meta learning.

Meta learning with multiple objectives thus has drawn much attention in recent studies. Specifically, some works study specific meta learning problems in the multi-objective case, such as multi-objective NAS [66, 57, 1, 34], multi-objective RL [4], and so on. However, those works either linearly combine multiple objectives into a single objective for the upper-level subproblem [65, 63, 11] or utilize multi-objective bi-level evolutionary algorithms [5, 54, 47] to handle it. The former approach needs to tune weights associated with all the objectives, which is time consuming, and its performance depends on the set of candidate weights in, for example, the cross validation method. The latter approach, whose computational complexity is even higher, has no convergence guarantee in the optimization process and is not easy to be integrated into gradient-based learning models such as deep neural networks, which limits its use in many learning models.

To alleviate those limitations in existing works, in this paper we propose a unified gradient-based Multi-Objective Meta Learning (MOML) framework with a convergence guarantee. The MOML framework formulates objective functions in meta learning with multiple objectives as a Multi-Objective Bi-Level optimization Problem (MOBLP), where the lower-level subproblem is to learn the adaptation to a task similar to vanilla meta learning and the upper-level subproblem minimizes a vector-valued function corresponding to multiple objectives for the meta learner. To solve MOBLP, we devise the first gradient-based optimization algorithm by alternatively solving the lower-level and upper-level subproblems via a gradient descent method and a gradient-based multi-objective optimization method such as [7], respectively. We theoretically prove the convergence properties of the proposed gradient-based optimization method. To show the effectiveness of the MOML framework, we apply it to several meta learning problems, including few-shot learning, domain adaptation, multi-task learning, and NAS, where multi-task learning is firstly formulated from the perspective of meta learning. In summary, the main contributions of this paper are four-fold.

- We propose a unified MOML framework based on the MOBLP and devise a gradient-based optimization algorithm for the MOML framework.

- We prove the convergence property of the proposed optimization algorithm.

- We formulate several learning problems as instances of the MOML framework.

- Experiments show that MOML achieves state-of-the-art performance on those learning problems.

## 2 Related Work

**Meta Learning**. Meta learning (a.k.a. learning to learn) learns knowledge from multiple tasks and then adapts it to new tasks with a few samples quickly. Many studies in meta learning mainly focus on solving the few-shot learning problem. From this view, meta learning can be divided into three main categories, including metric-based approach [55, 56], model-based approach [28], and optimization-based approach [12, 39]. For example, as an optimization-based method, MAML learns an initialization of model parameters so that a new task can be learned with a few training samples by fine-tuning the learned initialization. A widely-used formulation in meta learning can be cast as a bi-level optimization problem, where the upper-level subproblem is to learn meta parameters by minimizing a meta objective and the lower-level subproblem is to quickly adapt to new tasks with meta parameters [45, 19]. For example, MAML adapts to a new task by using the associated training dataset and the learned initialization in the lower level, and then updates the initialization according to the validation performance in the upper level. From this perspective, meta learning is a general

learning paradigm and has more general applications [19]. In this paper, we study meta learning from the perspective of the bi-level optimization.

**Multi-Objective Optimization**. Multi-objective optimization is to address the problem of simultaneously minimizing multiple objectives, which may conflict with each other. The simplest way to handle multi-objective optimization is to convert to a single-objective optimization problem via the linear scalarization approach which minimizes the weighted sum of all the objectives. Actually, many machine learning algorithms adopt this approach. For example, minimizing the regularization term to control the model complexity and minimizing the training loss are two conflicting objectives, and a learning model is usually to minimize a linear combination of those two terms. Recently, many kinds of advanced multi-objective optimization algorithms are proposed, such as evolutionary algorithms [73], population-based algorithms [15], gradient-based algorithms [7, 37], and so on. Some of them have been successfully applied to solve machine learning problems [22]. In this work, we focus on gradient-based algorithms because this approach can be easily integrated into gradient-based machine learning models such as deep learning models.

# 3 The MOML Framework

In this paper, the proposed MOML framework has a unified objective function, which is formulated as a **Multi-Objective Bi-Level optimization Problem** (MOBLP), as

$$\min_{\alpha \in \mathcal{A}, \omega \in \mathbb{R}^p} F(\omega, \alpha) = (F_1(\omega, \alpha), F_2(\omega, \alpha), ..., F_m(\omega, \alpha))^T \ \text{s.t.} \ \omega \in \mathcal{S}(\alpha), \tag{1}$$

where function $F : \mathbb{R}^p \times \mathbb{R}^n \to \mathbb{R}^m$ is a vector-valued jointly continuous function for the $m$ desired meta objectives and $\mathcal{A}$ is a nonempty compact subset of $\mathbb{R}^n$. In problem (1), $\mathcal{S}(\alpha)$ is defined as the set of optimal solutions to minimize $f(\omega, \alpha)$ *w.r.t.* $\omega$, *i.e.*,

$$\mathcal{S}(\alpha) = \arg\min_{\omega} f(\omega, \alpha). \tag{2}$$

When $m$ equals 1, problem (1) reduces to a standard Bi-Level optimization Problem (BLP), which is a widely-used formulation in meta learning [19], and hence from this perspective, the MOML framework is a generalization of meta learning. In problems (1) and (2), $F$ is called the Upper-Level (UL) subproblem and $f : \mathbb{R}^p \times \mathbb{R}^n \to \mathbb{R}$ is the Lower-Level (LL) subproblem. The LL subproblem can be considered as a constraint to the UL subproblem. In MOML, $F$ contains multiple meta objectives to be achieved for the meta learner and $f$ defines the objective function for current task such as the training loss. In Section 5, we will see the application of MOML in different learning problems, including few-shot learning, domain adaptation, multi-task learning, and neural architecture search.

To solve problem (1), there exist a type of works [5, 54, 47] which adopt multi-objective evolutionary algorithms. However, such methods have a high complexity without convergence guarantee and are not easy to be integrated with gradient-based models such as deep neural networks. Hence, we do not include them in experiments. To the best of our knowledge, there is no gradient-based optimization algorithm with convergence guarantee to solve an MOBLP, which is what we will do in the next section.

# 4 Optimization

In this section, we devise a general algorithm to solve the MOBLP (*i.e.*, problem (1)) and provide convergence analyses under certain assumptions.

## 4.1 Lower-Level Singleton Condition

Due to the complicated dependency between UL and LL variables, solving the MOBLP is challenging, especially when optimal solutions of the LL subproblem are not unique.

For a BLP with a single objective in the UL subproblem, many studies [8, 13, 52] potentially require that the LL subproblem only admits a unique minimizer for every $\alpha \in \mathcal{A}$. This condition can simplify both the optimization process and convergence analyses and is first fomally introduced as Lower-Level Singleton (LLS) condition in [30].

For the MOBLP, the LLS condition is necessary. If the LLS condition does not hold, the MOBLP is even ill-defined [9]. To see this, suppose for a fixed $\alpha_0$, we get a set of solutions $S(\alpha_0)$ for the LL subproblem. Since $F$ is vector-valued in problem (1), it is unclear that at which $\omega \in S(\alpha_0)$ the UL subproblem $F$ should be evaluated. With the LLS condition, problem (1) can be simplified as

$$\min_{\alpha \in \mathcal{A}} \varphi(\alpha) = F(\omega^*(\alpha), \alpha) \quad \text{s.t.} \quad \omega^*(\alpha) = \arg\min_{\omega} f(\omega, \alpha). \tag{3}$$

## 4.2 Gradient-based Optimization Algorithm

Here we design a gradient-based optimization algorithm to solve problem (3). Usually, there is no closed form for the solution $\omega^*(\alpha)$ of the LL subproblem and so it is difficult to optimize the UL subproblem directly. Another approach is to use the optimality condition of the LL subproblem (*i.e.*, $\nabla_\omega f(\omega, \alpha) = 0$) as equality constraints for the UL subproblem in a way similar to [41]. However, this approach only works for LL subproblems with simple forms and cannot work for general learning models.

To solve problem (3), we take a strategy similar to the alternating optimization. In the first part of each iteration (corresponding to steps 3-6 in Algorithm 1), we solve the LL subproblem via gradient descent methods. Specifically, with an initialization $\omega_0$ for the LL variable where the index $t$ of the iteration is omitted for notation simplicity, the solution of the LL subproblem can be updated for $K$ steps as $\omega_{k+1}(\alpha) = \mathcal{T}_k(\omega_k(\alpha), \alpha)$, $k = 1, \ldots, K - 1$, where $\mathcal{T}_k$ represents an operator to update $\omega$. Here we consider a first-order gradient descent method for $\mathcal{T}_k$ such as the Stochastic Gradient Descent (SGD) method and $\mathcal{T}_k$ can be formulated explicitly as $\mathcal{T}_k(\omega_k(\alpha), \alpha) = \omega_k(\alpha) - \mu \nabla_\omega f(\omega_k(\alpha), \alpha)$, where $\mu > 0$ denotes the step size and $\nabla_\omega f(\omega_k(\alpha), \alpha)$ denotes the derivative of $f$ *w.r.t.* $\omega$ at $\omega = \omega_k(\alpha)$. In the second part of each iteration (corresponding to steps 7-9 in Algorithm 1), by fixing the value of $\omega$ as the current solution $\omega_K(\alpha)$ obtained in the first part, we solve the UL subproblem as

$$\min_{\alpha} \varphi_K(\alpha) = F(\omega_K(\alpha), \alpha). \tag{4}$$

Problem (4) is an unconstrained Multi-Objective optimization Problem (MOP) and we can use any multi-objective optimization algorithms to solve it. Here we choose gradient-based multi-objective optimization methods as they can be seamlessly integrated into any gradient-based learning framework. There are several gradient-based multi-objective optimization algorithms [44, 7, 58] and they commonly find an appropriate descent direction $d$ for all the objectives in $F$ by aggregating their gradients *w.r.t.* $\alpha$. So

---

**Algorithm 1** Optimization Algorithm for MOML

**Input:** numbers of iterations $(T, K)$, step size $(\mu, \nu)$
1: Randomly initialized $\alpha_0$;
2: **for** $t = 1$ **to** $T$ **do**
3:     Initialize $\omega_0^t(\alpha_t)$;
4:     **for** $j = 1$ **to** $K$ **do**
5:         $\omega_j^t(\alpha_t) \leftarrow \omega_{j-1}^t(\alpha_t) - \mu \nabla_\omega f(\omega_{j-1}^t(\alpha_t), \alpha_t)$;
6:     **end for**
7:     Compute gradients $\nabla_\alpha F_i(\omega_K^t(\alpha_t), \alpha_t)$ for all the $i$'s;
8:     Compute the gradient
    $d(\omega_K^t(\alpha_t), \alpha_t) = \text{MOPSolver}(\{\nabla_\alpha F_i(\omega_K^t(\alpha_t), \alpha_t)\})$;
9:     $\alpha_{t+1} = \alpha_t - \nu d(\omega_K^t(\alpha_t), \alpha_t)$ with a step size $\nu$;
10: **end for**

---

such process is denoted by $d = \text{MOPSolver}(\{\nabla_\alpha F_i(\omega_K(\alpha), \alpha)\}_{i=1}^m)$ in Algorithm 1. In this paper, we adopt a simple gradient-based MOP method called Multiple Gradient Descent Algorithm (MGDA) [7], whose details are introduced in Appendix A.2 due to page limit.

The entire algorithm to solve problem (3) is shown in Algorithm 1, which to the best of our knowledge is the first gradient-based optimization algorithm for MOBLPs. In Algorithm 1, we obtain only one solution for MOBLPs, which is different from multi-objective evolutionary algorithms that can find a population of solutions. How to obtain multiple nearly optimal solutions for MOBLPs is beyond the scope of this paper and we will study it in our future work.

## 4.3 Convergence Analysis

Algorithm 1 is simple and intuitive, but its convergence properties are unknown, which is what we aim to study in this section. As an MOBLP, problem (3) cannot reduce to SOML with a scalar-valued objective function in the upper-level subproblem when using gradient-based Algorithm 1 to solve it. Therefore, it has different theoretical properties from SOML as we need to focus on the convergence properties of a minimal point set instead of a minimum scalar in the UL subproblem.

We first recall some notions about vector-valued functions. Consider a vector-valued function $g(z) : \mathbb{R}^n \to \mathbb{R}^m$ $(m, n \in \mathbb{N}, m \geq 2)$. We denote by $\mathrm{Min}\, g(z)$ the set of all the minimal points of function $g(z)$. $\mathrm{Min}\, g(z)$ is also called the Pareto frontier or Pareto-optimal set. The corresponding efficient solution or Pareto-optimal solution set of $g(z)$ is denoted by $\mathrm{Eff}(g(z))$. The convexity of vector-valued functions is called the P-convex. The details of these definitions can be found in Appendix A.

To help analyze convergence properties of Algorithm 1, with a basic assumption about the LLS condition which is mentioned in Section 4.1 and is widely adopted in BLPs [52, 10], we can obtain the following result.

**Theorem 1.** *If $f(\omega, \alpha)$ is jointly continuous, $\arg\min_\omega f(\omega, \alpha)$ is a singleton for every $\alpha \in \mathcal{A}$, and $\omega^*(\alpha)$ is uniformly bounded on $\mathcal{A}$, then the function $F(\omega^*(\alpha), \alpha)$ is continuous w.r.t. $\alpha$.*

Because $\mathcal{A}$ is a compact set, Theorem 1 implies the existence of solutions. Theorem 1 and the uniform convergence of $\omega_K(\alpha)$ can further imply the convergence for the solution of the LL subproblem, which is similar to that of the standard BLP problem [13].

For the convergence of Algorithm 1, we need to analyze minimal point sets of the images of perturbed functions $\varphi_K(\alpha)$ and $\varphi(\alpha)$ which are defined in problems (4) and (3), respectively. We consider the most natural set convergence under this setting, *i.e.*, the Kuratowski-Painlevé set-convergence. Due to page limit, please refer to these definitions in Appendix A. Under certain assumptions which are used in analyses of BLP and MOP [13, 35], we have the following convergence results.

**Theorem 2.** *In addition to Theorem 1, it is assumed that $(i)$ $F(\cdot, \alpha)$ is uniformly Lipschitz continuous; $(ii)$ The iterative sequence $\{\omega_k(\alpha)\}_{k=1}^K$ converges uniformly to $\omega^*(\alpha)$ on $\mathcal{A}$ as $K \to +\infty$; $(iii)$ $\mathcal{A}$ is a convex set; $(iv)$ $\varphi_K$ is P-convex and $\varphi$ is strictly P-convex. Then, the Kuratowski-Painlevé set-convergence of both the minimal point set and efficient solution set in Algorithm 1 holds, i.e.,*

$$\mathrm{Min}\, \varphi_K(\alpha) \to \mathrm{Min}\, \varphi(\alpha), \ \mathrm{Eff}\, \varphi_K(\alpha) \to \mathrm{Eff}\, \varphi(\alpha).$$

Thus, under some specific assumptions, Theorem 2 provides a theoretical justification for the convergence of Algorithm 1 to solve MOBLPs.

## 5 Applications of MOML

In this section, we introduce several use cases of the MOML framework in several learning problems, including few-shot learning, semi-supervised domain adaptation, multi-task learning, and neural architecture search.

### 5.1 Few-Shot Learning

Few-Shot Learning (FSL) aims to tackle the problem of training a model with only a few training samples [64]. Recently, FSL is widely studied from the perspective of meta learning by using prior knowledge in the meta training process. Most studies in FSL only consider the classification performance. However, in real-world applications, the performance is not the only important focus. For example, we expect FSL models to not only have good performance but also be robust to adversarial attacks [26], which may improve the generalization of FSL models. In the following, we can see that this setting can naturally be modeled by the proposed MOML framework.

**Problem Formulation.** Suppose there is a base dataset $\mathcal{D}_{base}$ with a category set $\mathcal{C}_{base}$ and a novel dataset $\mathcal{D}_{novel}$ with a category set $\mathcal{C}_{novel}$, where $\mathcal{C}_{base} \cap \mathcal{C}_{novel} = \emptyset$. The goal of FSL is to adapt the knowledge learned from $\mathcal{D}_{base}$ to help the learning of $\mathcal{D}_{novel}$. In the $i$th meta training episode, we generate from $\mathcal{D}_{base}$ an $N$-way $k$-shot classification task, which consists of a support set $\mathcal{D}_{base}^{s(i)}$ and a query set $\mathcal{D}_{base}^{q(i)}$. For the robustness, we add perturbations generated by the Projected Gradient Descent (PGD) method [26] into each data point in $\mathcal{D}_{base}^{q(i)}$ to generate a perturbed query set $\mathcal{D}_{base}^{q(i),adv}$. The objective function of the FSL model that considers both the performance and the robustness can be formulated as

$$\min_\alpha \ \left(\mathcal{L}_F(\omega^{*(i)}(\alpha), \alpha, \mathcal{D}_{base}^{q(i)}), \mathcal{L}_F(\omega^{*(i)}(\alpha), \alpha, \mathcal{D}_{base}^{q(i),adv})\right)$$

$$\text{s.t. } \omega^{*(i)}(\alpha) = \arg\min_\omega \mathcal{L}_F(\omega, \alpha, \mathcal{D}_{base}^{s(i)}), \tag{5}$$

where $\omega$ represents model parameters, $\alpha$ denotes the meta parameters to encode common knowledge that can be transferred to novel tasks, and $\mathcal{L}_F(\omega, \alpha, \mathcal{D})$ denotes the average classification loss of a model with model parameters $\alpha$ and meta parameters $\omega$ on a dataset $\mathcal{D}$. In the UL subproblem of problem (5), the first objective measures the classification loss on the query set based on $\omega^{*(i)}(\alpha)$ obtained by solving the LL subproblem and the second objective measures the robustness via the classification performance on the perturbed query set. Problem (5) provides a general formulation, which depends on what $\alpha$ represents, for FSL. We adapt problem (5) to three FSL methods (*i.e.*, MAML [12], ProtoNet [55] and BOIL [40]) and the detailed formulations are put in Appendix C. It is easy to see that problem (5) fits the MOML framework and we can use Algorithm 1 to solve it.

For an MOBLP such as problem (1), a common approach is to transform the UL subproblem into a single objective problem via the linear scalarization approach [65, 63, 11]. In contrast with MOML, this approach is named as Single-Objective Meta Learning (SOML) in this paper and used as an extra baseline method in our experiments. Specifically, with fixed weights $\{\gamma_i\}_{i=1}^m$, which satisfy $\sum_{i=1}^m \gamma_i = 1$ and $\gamma_i \geq 0$, problem (1) is transformed to a standard bi-level optimization problem as $\min_{\alpha, \omega \in \mathcal{S}(\alpha)} \sum_{i=1}^m \gamma_i F_i(\omega, \alpha)$. This approach heavily depends on the selection of weights $\{\gamma_i\}_{i=1}^m$ and inappropriate weights usually lead to poor performance.

**Experiments**. Experiments are conducted on two FSL benchmark datasets, *mini*-ImageNet [61] and CUB-200-2011 (referred to as CUB) [62]. *Due to page limit, experimental settings and experimental results on the CUB dataset are put in Appendix C.* The original baseline methods (*i.e.*, MAML, ProtoNet, and BOIL) are trained in a conventional way and we evaluate the clean accuracy and PGD accuracy of all the methods, where the clean accuracy is measured on the original test query set and the PGD accuracy is tested on the query set perturbed by the PGD attack. As the clean accuracy and the PGD accuracy are reported to be conflicting to each other [70], inspired by the widely-used F1-score based on the precision and recall, we propose a new metric called Balance score (B-score), which is defined as B-score $= 2 \times (\text{CA} \times \text{PA})/(\text{CA} + \text{PA})$ with CA and PA denoting the clean accuracy and PGD accuracy, respectively, to fully measure the performance in terms of the clean accuracy and PGD accuracy.

Table 1: Classification accuracy (abbreviated as "Clean Acc.") and PGD accuracy (abbreviated as "PGD Acc.") on the *mini*-ImageNet dataset for 5-way $k$-shot FSL. The best result in each group of methods is highlighted in **bold** and the best result in each setting is annotated with underline.

| | Method | Clean Acc. | PGD Acc. | B-score |
|---|---|---|---|---|
| **1-shot** | MAML [12] | 45.24±0.81 | 1.18±0.15 | 2.10±0.52 |
| | MAML+SOML | 40.78±0.75 | 23.91±0.67 | 29.83±0.43 |
| | MAML+MOML (**ours**) | 39.23±0.76 | 25.80±0.67 | **31.12**±0.70 |
| | ProtoNet [55] | 44.67±0.75 | 2.60±0.21 | 3.73±0.35 |
| | ProtoNet+SOML | 38.65±0.72 | 23.10±0.65 | 28.67±0.67 |
| | ProtoNet+MOML (**ours**) | 35.06±0.70 | 27.24±0.65 | **30.51**±0.66 |
| | BOIL [40] | 47.64±0.85 | 3.05±0.22 | 5.53±0.61 |
| | BOIL+SOML | 40.44±0.79 | 25.94±0.69 | 31.29±0.75 |
| | BOIL+MOML (**ours**) | 41.22±0.83 | 27.77±0.75 | **32.98**±0.79 |
| **5-shot** | MAML [12] | 61.88±0.77 | 2.82±0.21 | 5.01±0.43 |
| | MAML+SOML | 56.16±0.72 | 34.85±0.72 | 42.91±0.71 |
| | MAML+MOML (**ours**) | 55.66±0.78 | 39.38±0.77 | **45.89**±0.77 |
| | ProtoNet [55] | 66.55±0.70 | 0.68±0.09 | 1.31±0.18 |
| | ProtoNet+SOML | 59.11±0.71 | 39.41±0.73 | 46.93±0.71 |
| | ProtoNet+MOML (**ours**) | 58.72±0.74 | 41.59±0.75 | **48.59**±0.74 |
| | BOIL [40] | 66.02±0.72 | 4.85±0.30 | 1.11±0.51 |
| | BOIL+SOML | 58.54±0.76 | 34.28±0.75 | 42.94±0.78 |
| | BOIL+MOML (**ours**) | 60.21±0.79 | 35.47±0.78 | **44.37**±0.78 |

The average results over 600 testing tasks on the *mini*-ImageNet dataset are presented in Table 1. From the results, we can see that the adversarial robustness of the original FSL methods (*i.e.*, MAML, ProtoNet, and BOIL) is poor, while SOML and the proposed MOML can significantly improve the PGD accuracy. For example, MOML can improve the PGD accuracy of ProtoNet by about 40.91% under the 5-shot setting. Since the classification accuracy and adversarial robustness are conflicting [70], the clean accuracy of SOML and MOML slightly drops when compared with the orignal FSL baselines. The B-score of MOML is higher than SOML, which indicates the multi-objective formulation in the UL subproblem of MOML is better than the single-objective formulation in SOML.

## 5.2 Semi-Supervised Domain Adaptation

Semi-Supervised Domain Adaptation (SSDA) aims to address the domain shift between two domains so that the model trained in a label-rich source domain can be adapted to a target domain with limited labeled samples and abundant unlabeled samples [69]. A widely used approach for SSDA is to align the distributions of two domains by finding some domain-invariant components. There are usually three objectives to be considered, including two classification losses on two domains and an alignment loss to measure the domain discrepancy. While existing works such as [49, 74] optimize

all the objectives by simply computing a weighted sum of them, we formulate the SSDA problem as an MOBLP under the MOML framework.

**Problem Formulation**. Given a source domain $\mathcal{S}$ and a target domain $\mathcal{T}$, the source domain has a large labeled dataset $\mathcal{D}_{\mathcal{S}}$ and the target domain has a limited labeled dataset $\mathcal{D}_{\mathcal{T}}^l$ as well as a large unlabeled dataset $\mathcal{D}_{\mathcal{T}}^u$, where $\mathcal{D}_{\mathcal{T}} = \mathcal{D}_{\mathcal{T}}^l \bigcup \mathcal{D}_{\mathcal{T}}^u$ denotes the entire dataset for the target domain. The average classification losses in the source and target domains are represented by $\mathcal{L}_D(\omega, \mathcal{D}_{\mathcal{S}})$ and $\mathcal{L}_D(\omega, \mathcal{D}_{\mathcal{T}}^l)$, respectively, where $\omega$ is the model parameter. The alignment loss denoted by $\mathcal{L}_A(\omega, \alpha, \mathcal{D}_{\mathcal{S}}, \mathcal{D}_{\mathcal{T}}^u)$ aims to learn domain-invariant components such as a domain-invariant projection space by minimizing the local maximum mean discrepancy in DSAN [74] or domain-invariant prototypes by maximizing the entropy in MME [49], where $\alpha$ is the initialization of $\omega$. Then we can formulate the SSDA problem under the MOML framework as

$$\min_{\alpha} \ (\mathcal{L}_D(\omega^*(\alpha), \mathcal{D}_{\mathcal{S}}), \mathcal{L}_D(\omega^*(\alpha), \mathcal{D}_{\mathcal{T}}^l), \mathcal{L}_A(\omega, \alpha, \mathcal{D}_{\mathcal{S}}, \mathcal{D}_{\mathcal{T}}^u))$$
$$\text{s.t.} \ \ \omega^*(\alpha) = \arg\min_{\omega} \mathcal{L}_A(\omega, \alpha, \mathcal{D}_{\mathcal{S}}, \mathcal{D}_{\mathcal{T}}^u). \tag{6}$$

In the LL subproblem of problem (6), we aim to learn a model to find a domain-invariant component between two domains via the alignment loss $\mathcal{L}_A$ by optimizing $\omega$ with an initialization $\alpha$, and in the UL subproblem, we expect to improve the model further by updating $\alpha$ via minimizing the two classification losses together with the alignment loss. Here $\alpha$ acts similar to the parameter initialization in MAML (*i.e.*, $\alpha$ in problem (5)) and helps learn $\omega$ in the LL subproblem, but it does not require any adaptation on the testing process. We can follow two state-of-the-art domain adaptation models (*i.e.*, DSAN and MME) to design the alignment loss and due to page limit, the detailed formulation of the alignment loss $\mathcal{L}_A$ is put in Appendix D.

Although a bi-level objective function is also formulated in Meta-MME [27], there exist two significant differences between Meta-MME and the proposed MOML method. Firstly, Meta-MME aims to learn the initialization of the network parameters by minimizing the source classification loss and the alignment loss in its LL subproblem and then validate it on a few target labeled samples in the UL subproblem, which is different from the proposed MOML formulation (*i.e.*, problem (6)). Secondly, the proposed MOML method considers a multi-objective optimization problem in the UL subproblem, which is different from Meta-MME. Empirically, MOML outperforms the Meta-MME method as shown in Table 2.

Table 2: Classification accuracy (%) on the Office-31 dataset with ResNet-50 as backbone for SSDA. † means the corresponding method is appropriately modified to adapt to our experimental setting according to the released code. ‡ indicates that the corresponding model is reimplemented by us. ↑, ↓, and − in the subscript indicate an increase, a decrease, and no change, respectively, when compared with the original method in two groups of models based on DSAN or MME. The best results in each group of models (*i.e.*, DSAN and MME) are highlighted in **bold** and the best results of each transfer task are annotated with underlines.

| Method | A→D | D→A | A→W | W→A | D→W | W→D | Avg |
|---|---|---|---|---|---|---|---|
| S+T | 93.58 | 74.16 | 92.17 | 74.08 | 98.01 | 100 | 88.67 |
| DANN[†] [14] | 93.09 | 74.52 | 91.60 | 75.07 | 98.58 | 100 | 88.81 |
| ENT[†] [16] | 93.33 | 74.16 | 94.44 | 75.03 | 97.86 | 100 | 89.13 |
| APE[†] [23] | 94.81 | 76.10 | 91.80 | 76.02 | 97.29 | 99.75 | 89.29 |
| ADR[†] [50] | 93.33 | 76.73 | 93.30 | 76.51 | 97.86 | 100 | 89.62 |
| CDAN[†] [32] | 94.32 | 75.25 | 92.88 | 76.98 | 98.43 | 100 | 89.64 |
| DSAN[†] [74] | 93.83 | 76.82 | 93.59 | 75.68 | **98.43** | **100** | 89.73 |
| DSAN+SOML | **94.32**$_\uparrow$ | 76.91$_\uparrow$ | 94.16$_\uparrow$ | **75.99**$_\uparrow$ | 97.72$_\downarrow$ | **100**$_-$ | 89.85$_\uparrow$ |
| DSAN+MOML (**ours**) | 94.08$_\uparrow$ | **77.13**$_\uparrow$ | **94.59**$_\uparrow$ | 75.96$_\uparrow$ | 98.36$_\downarrow$ | **100**$_-$ | **90.02**$_\uparrow$ |
| MME[†] [49] | 92.09 | 77.71 | 93.73 | 77.27 | 98.01 | **100** | 89.80 |
| Meta-MME[‡] [27] | 92.10 | 77.01 | 94.30 | 76.87 | 98.29 | **100** | 89.76 |
| MME+SOML | 92.09$_-$ | 77.57$_\downarrow$ | 94.30$_\uparrow$ | 77.20$_\downarrow$ | 98.29$_\uparrow$ | **100**$_-$ | 89.90$_\uparrow$ |
| MME+MOML (**ours**) | **94.32**$_\uparrow$ | **78.30**$_\uparrow$ | 94.44$_\uparrow$ | **77.71**$_\uparrow$ | 98.43$_\uparrow$ | **100**$_-$ | **90.53**$_\uparrow$ |

**Experiments**. Experiments are conducted on the Office-31 dataset [48], which has 3 domains: Amazon (**A**), Webcam (**W**) and DSLR (**D**). By following [59, 33], we construct all six transfer

tasks. Each class in the target domain has three labeled images in the training process by following [49]. Baseline models in comparison include a deep neural network (denoted by 'S+T') that is trained on $\mathcal{D}_\mathcal{S} \bigcup \mathcal{D}_\mathcal{T}^l$ and other eight state-of-the-arts domain adaptation methods: **ENT** [16], **DANN** [14], **ADR** [50], **CDAN** [32], **MME** [49], **Meta-MME** [27], **APE** [23] and **DSAN** [74]. As DSAN and MME are two state-of-the-art domain adaptation models, to improve their performance further, SOML and MOML adopt the alignment loss in DSAN and MME, respectively, leading to two groups of models, *i.e.*, (DSAN, DSAN+SOML, DSAN+MOML) and (MME, Meta-MME, MME+SOML, MME+MOML). All the methods except S+T are trained on $\mathcal{D}_\mathcal{S} \bigcup \mathcal{D}_\mathcal{T}$. *Due to page limit, details of baselines and experimental settings are put in Appendix D.*

Experimental results on the Office-31 dataset are shown in Table 2. The incorporation of SOML into DSAN and MME can consistently improve the performance of two baseline models and the proposed MOML method further improves the classification accuracy. For example, MME+MOML significantly improves the performance of the first five tasks and achieves the perfect performance (*i.e.*, the 100% accuracy) on the last task as most baselines did. Moreover, compared with state-of-the-art baselines, DSAN+MOML achieves the best result (*i.e.*, 94.59%) on transfer task A→W and MME+MOML has the best accuracy (*i.e.*, 78.30% and 77.71%) on transfer tasks D→A and W→A. Besides, MME+MOML achieves the best average classification accuracy of 90.53% and significantly outperforms all of the baselines, which indicates the effectiveness of the proposed MOML method.

### 5.3 Multi-Task Learning

Multi-Task Learning (MTL) [2, 72] aims to improve the performance of multiple tasks simultaneously by leveraging useful information contained in these tasks. Learning the loss weighting is a challenge in MTL and there are some works [51, 31, 71] to solve this problem. Among those works, Sener and Koltun [51] formulate multi-task learning problems from the perspective of multi-objective optimization and implicitly learn the task weights via MGDA, Liu *et al.* [31] estimate the task weight of each task as the ratio of the training losses in the last two iterations for the corresponding task, and Yu *et al.* [71] project each task's gradient onto the normal plane of the other. Different from those works which are all based on single-level optimization problems on the entire training set, for the first time we formulate this problem as an MOBLP based on the split of the entire training dataset and solve this problem based on the MOML framework.

**Problem Formulation**. Suppose there are $m$ tasks. The $i$th task has a dataset $\mathcal{D}_i$ for model training. Here each $\mathcal{D}_i$ is partitioned into two subsets: the training dataset $\mathcal{D}_i^{tr}$ and the validation dataset $\mathcal{D}_i^{val}$, where $\mathcal{D}_i^{tr}$ is used to train a multi-task model and $\mathcal{D}_i^{val}$ is to measure the performance of the multi-task model on the $i$th task. $f(\cdot; \omega)$, the learning function of the multi-task model parameterized by $\omega$, receives data points from the $m$ tasks and outputs predictions. Let $\alpha_i \in [0, 1]$ denotes the loss weight for the $i$th task. The goal is to jointly learn the loss weights $\boldsymbol{\alpha} = (\alpha_1, \ldots, \alpha_m)^T$ and the model parameter $\omega$. The objective function of the proposed method under the MOML framework is formulated as

$$\min_{\boldsymbol{\alpha}} \left( \mathcal{L}_{MTL}(\omega^*(\boldsymbol{\alpha}), \mathcal{D}_1^{val}), \ldots, \mathcal{L}_{MTL}(\omega^*(\boldsymbol{\alpha}), \mathcal{D}_m^{val}) \right)$$

$$\text{s.t. } \omega^*(\boldsymbol{\alpha}) = \arg\min_{\omega} \sum_{i=1}^{m} \alpha_i \mathcal{L}_{MTL}(\omega, \mathcal{D}_i^{tr}), \ 0 \leq \alpha_i \leq 1 \ \forall i, \sum_i^m \alpha_i = 1, \tag{7}$$

where $\mathcal{L}_{MTL}(\omega, \mathcal{D}) = \frac{1}{|\mathcal{D}|} \sum_{(\mathbf{x},y) \in \mathcal{D}} \ell(f(\mathbf{x}; \omega), y)$ denotes the average loss of $f(\cdot; \omega)$ on a dataset $\mathcal{D}$ with $|\mathcal{D}|$ denoting the size of $\mathcal{D}$ and $\ell(\cdot, \cdot)$ denoting a loss function. In the LL subproblem of problem (7), when given weights in $\boldsymbol{\alpha}$, we aim to learn a MTL model to get optimal parameters $\omega^*$ on the training dataset and in the UL subproblem, we expect to update $\boldsymbol{\alpha}$ via minimizing the loss of the trained MTL model with parameters $\omega^*$ on the validation dataset of each task.

**Experiments**. Experiments are conducted on the NYUv2 [53], Office-31 and Office-Home [60] datasets. Baseline methods in the comparison include different loss weighting strategies such as Equal Weights (**EW**), Dynamic Weight Average (**DWA**) [31] with the temperature parameter $T$ as 2, **MGDA** [51], **PCGrad** [71] and **SOML**, and they are built on two multi-task architectures, including Deep Multi-Task Learning (**DMTL**) that adopts the hard-sharing structure to share the first several layers and Multi-Task Attention Network (**MTAN**) [31]. The ResNet-50 is used as the backbone and we do not use data augmentation. *Due to page limit, the introduction of datasets, details on*

*experimental settings, and experimental results on the Office-31 and Office-Home datasets are put in Appendix E.*

Table 3: Performance on the NYUv2 dataset with three tasks: 13-class semantic segmentation, depth estimation, and surface normal prediction. The best combinations of the architecture and weighting strategy are highlighted in **bold**. The best results for each task on each measure are annotated with underlines. ↑ (↓) means the higher (lower) the result, the better the performance.

| Architecture | Weighting Strategy | Segmentation | | Depth | | Surface Normal | | | | |
|---|---|---|---|---|---|---|---|---|---|---|
| | | | | | | Angle Distance | | Within $t°$ | | |
| | | mIoU↑ | Pix Acc ↑ | Abs Err ↓ | Rel Err↓ | Mean ↓ | Median ↓ | 11.25 ↑ | 22.5 ↑ | 30 ↑ |
| DMTL | EW | 52.71 | 74.78 | 0.3886 | **0.1581** | 23.8568 | 17.3537 | 34.57 | 60.45 | 71.63 |
| | DWA [31] | 52.72 | 75.11 | 0.3931 | 0.1631 | 23.7894 | 17.3320 | 34.57 | 60.51 | 71.67 |
| | MGDA [51] | 52.89 | 74.87 | 0.3963 | 0.1638 | 23.7513 | 17.2685 | 34.74 | 60.64 | 71.75 |
| | PCGrad [71] | 53.22 | 75.45 | 0.3920 | 0.1658 | 23.2904 | 16.8728 | 35.47 | 61.58 | 72.56 |
| | SOML | 53.20 | 75.22 | 0.3923 | 0.1628 | 23.6885 | 17.0228 | 35.42 | 61.03 | 71.97 |
| | MOML (**ours**) | 54.98 | **75.98** | **0.3877** | 0.1618 | **23.2401** | **16.7388** | **35.90** | **61.81** | **72.76** |
| MTAN [31] | EW | 53.97 | 75.90 | 0.3794 | 0.1580 | 22.8743 | 16.5502 | 36.54 | 62.52 | 73.31 |
| | DWA [31] | 54.12 | 75.79 | 0.3902 | 0.1595 | 22.9691 | 16.6212 | 36.23 | 62.41 | 73.28 |
| | MGDA [51] | 54.38 | 75.55 | 0.3854 | 0.1583 | 22.9396 | 16.4670 | 36.70 | 62.58 | 73.23 |
| | PCGrad [71] | **54.40** | **76.13** | 0.3830 | 0.1581 | 23.0040 | 16.4636 | 36.67 | 62.65 | 73.34 |
| | SOML | 54.03 | 75.48 | 0.3829 | 0.1581 | 22.8279 | 16.4259 | 36.74 | 62.67 | 73.46 |
| | MOML (**ours**) | 54.23 | 75.63 | 0.3843 | 0.1567 | **22.7530** | **16.2468** | 37.20 | **63.09** | 73.65 |

Experimental results on the NYUv2 dataset are shown in Table 3. Firstly, SOML outperforms the EW, DWA, and MGDA strategies when using the DMTL architecture and achieves comparable performance with the four baselines with the MTAN architecture. It indicates the proposed bi-level formulation (*i.e.*, problem (7) with weighted combined objectives in the UL subproblem used in SOML) can achieve very good performance when compared with the state-of-the-art baselines. Secondly, the MOML method outperforms the SOML method under both DMTL and MTAN architectures, which means the multi-objective formulation in the UL subproblem is better than the single-objective formulation. Finally, it is noticeable that the proposed MOML method achieves state-of-the-art results in many metrics. For example, when training with the DMTL architecture, MOML can achieve 54.98% in terms of the mIoU, which is significantly higher than other baselines even with the advanced MTAN architecture. MOML with the MTAN architecture achieves the best results on six metrics. Experimental results on the NYUv2, Office-31, and Office-Home datasets, the latter two of which have their results in Appendix E, show that the proposed MOML framework achieves state-of-the-art performance for learning loss weights in MTL.

The loss weights learned by the proposed MOML are $(0.3120, 0.3804, 0.3076)$ and $(0.2541, 0.4860, 0.2599)$ for the three tasks (*i.e.*, semantic segmentation, depth estimation, and surface normal prediction) when using the DMTL and MTAN architectures, respectively. It is interesting to find that when using different architectures, the weight of the depth estimation task is commonly the highest, while the other two tasks have similar weights.

### 5.4 Neural Architecture Search

NAS aims to design the architecture of neural networks in an automated way. In architecture design, we usually need to consider multiple factors. For example, we expect that the searched architecture has good performance, behaves robustly to noises, and consumes low resources. In this case, we formulate the NAS with multiple objectives as a case of MOML. *Due to page limit, the introduction of datasets, details on experimental settings, and experimental results are put in Appendix F.*

**Problem Formulation**. By following the DARTS method [29], in an operation space denoted by $\mathcal{O}$, each element is an operation function $o(\cdot)$ and each cell is a directed acyclic graph with $N$ nodes, where each node represents a hidden representation and each edge $(i, j)$ denotes a candidate operation $o(\cdot)$ with a probability $\alpha_o^{(i,j)}$. Therefore, $\boldsymbol{\alpha} = \{\alpha_o^{(i,j)}\}_{(i,j)\in\boldsymbol{E},o\in\mathcal{O}}$ is a representation of the neural architecture, where $\boldsymbol{E}$ denotes the set of all the edges in all the cells. The entire dataset is split into a training dataset denoted by $\mathcal{D}_{tr}$ and a validation dataset denoted by $\mathcal{D}_{val}$.

The multi-objective NAS considers three objectives: the classification accuracy, adversarial robustness and the number of parameters. We formulate three corresponding losses as $\mathcal{L}_N(\omega, \boldsymbol{\alpha}, \mathcal{D}_{val}), \mathcal{L}_N(\omega, \boldsymbol{\alpha}, \mathcal{D}_{val}^{adv})$, and $\mathcal{L}_{nop}(\boldsymbol{\alpha})$, where $\omega$ denotes all the model parameters in the

neural network and $\mathcal{D}_{val}^{adv}$ denotes the perturbed validation dataset by adding perturbations on each data point. $\mathcal{L}_N(\omega, \boldsymbol{\alpha}, \mathcal{D})$ denotes the average classification loss on a dataset $\mathcal{D}$ of a neural network with parameters $\omega$ and an architecture $\boldsymbol{\alpha}$,

$$\mathcal{L}_N(\omega, \boldsymbol{\alpha}, \mathcal{D}) = \frac{1}{n} \sum_{(\mathbf{x}, y) \in \mathcal{D}} \ell(\omega, \mathbf{x}, y),$$

where $\ell(\omega, \mathbf{x}, y)$ denotes the loss function for each sample and $n$ is the size of $\mathcal{D}$.

To formulate $\mathcal{L}_{nop}(\boldsymbol{\alpha})$, we denote by $n_o$ the number of parameters associated with an operation $o$ and by $N_{nop}(\boldsymbol{\alpha})$ the number of parameters in a searched architecture $\boldsymbol{\alpha}$. Then $N_{nop}(\boldsymbol{\alpha})$ can be computed by $N_{nop}(\boldsymbol{\alpha}) = \sum_{(i,j) \in \boldsymbol{E}} n^{(i,j)}$, where $n^{(i,j)}$ is the number of parameters of the searched operation on the edge $(i, j)$. As we determine the operation of each edge by selecting the one with the largest probability, hence we have $n^{(i,j)} = n_{\arg\max_{o \in \mathcal{O}} \alpha_o^{(i,j)}}$. As the $\arg\max$ operation is non-differentiable, we use the softmax function to approximate $N_{nop}(\boldsymbol{\alpha})$ as $\hat{N}_{nop}(\boldsymbol{\alpha}) = \sum_{(i,j) \in \boldsymbol{E}} \sum_{o \in \mathcal{O}} \frac{\exp(\alpha_o^{(i,j)})}{\sum_{o' \in \mathcal{O}} \exp(\alpha_{o'}^{(i,j)})} n_o$. Therefore, to search a network architecture with an expected size $L$, $\mathcal{L}_{nop}(\boldsymbol{\alpha})$ can be formulated as

$$\mathcal{L}_{nop}(\boldsymbol{\alpha}) = |\hat{N}_{nop}(\boldsymbol{\alpha}) - L|. \tag{8}$$

Finally, the overall formulation for the multi-objective NAS is formulated as

$$\min_{\boldsymbol{\alpha}} \ (\mathcal{L}_N(\omega^*(\boldsymbol{\alpha}), \boldsymbol{\alpha}, \mathcal{D}_{val}), \mathcal{L}_N(\omega^*(\boldsymbol{\alpha}), \boldsymbol{\alpha}, \mathcal{D}_{val}^{adv}), \mathcal{L}_{nop}(\boldsymbol{\alpha}))$$
$$\text{s.t.} \ \ \omega^*(\boldsymbol{\alpha}) = \arg\min_{\omega} \ \mathcal{L}_N(\omega, \boldsymbol{\alpha}, \mathcal{D}_{tr}). \tag{9}$$

In the LL subproblem of problem (9), when given the architecture $\boldsymbol{\alpha}$, we can train a model with optimal parameters $\omega$ on the training dataset and in the UL subproblem, we expect to update the architecture $\boldsymbol{\alpha}$ by making a trade-off among the validation loss, the adversarial robustness and the number of parameters. Obviously problem (9) matches the MOML framework. It is easy to see that the DARTS method is a special case of problem (9) when its UL subproblem contains the first objective only and hence problem (9) generalizes the DARTS method by considering two more objectives.

Compared with the NSGANetV2 method [34] that utilizes a multi-objective bi-level evolutionary algorithm, MOML is more efficient and has a convergence guarantee. Moreover, NSGANetV2 uses ensembled surrogate models to predict the accuracy of an architecture, which may incur a performance gap between the UL and LL subproblems. The LL subproblem of NSGANetV2 only chooses over 300 candidate architectures for evaluation with a supernet constructed for weight sharing, which may lead to suboptimal solutions.

## 6  Conclusions

As a generalization of meta learning based on the bi-level formulation, a simple MOML framework based on the multi-objective bi-level optimization is proposed in this paper. In the MOML framework, the upper-level subproblem takes multiple objectives of a learning problem into consideration. To solve the objective function of the MOML framework, a gradient-based optimization algorithm is proposed and the convergence property of this algorithm is studied. Moreover, several use cases of the MOML framework are investigated to demonstrate the effectiveness of the MOML framework. In our future work, we will apply the MOML framework to more learning problems.

## Acknowledgements

This work is supported by NSFC grant 62076118. We thank Dr. Jin Zhang for helpful discussion.

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
