# Appendix for "Multi-Objective Meta Learning"

## A    Review of Multi-Objective Optimization

### A.1    Notations and Terminologies

We first recall some definitions in Multi-Objective optimization Problems (MOP), including the definition of the minimality and convexity of vector-valued functions and Kuratowski-Painlevé set-convergence [36].

Let $P$ be the set of non-negative real vectors $\mathbb{R}_+^m = \{l \in \mathbb{R}^m : \forall l_i \geq 0\}$, where $l_i$ denote the $i$th entry in $l$. The interior $\text{int}P$ denotes the set of positive real vectors $\text{int}P = \{l \in \mathbb{R}^m : \forall l_i > 0\}$. $P$, which is a closed and convex cone, and the interior $\text{int}P$ induce a partial order for any two points in $\mathbb{R}^m$. That is, for any $l^1, l^2 \in \mathbb{R}^m$, we define

$$l^1 \leq l^2 \iff l^2 - l^1 \in P$$
$$l^1 < l^2 \iff l^2 - l^1 \in \text{int}P.$$

That is, for $l^1, l^2 \in \mathbb{R}^m$, the partial ordering $l^1 \leq l^2$ and $l^1 < l^2$ imply that $l_i^1 \leq l_i^2$ and $l_i^1 < l_i^2$ for all $i \in \{1, ..., m\}$, respectively. Given $l^1 \in \mathbb{R}^m$, we define $l^1 - P = \{l \in \mathbb{R}^m : l \leq l^1\}$ and $l^1 - \text{int}P = \{l \in \mathbb{R}^m : l < l^1\}$. We now recall the notions of minimality for a subset in $\mathbb{R}^m$.

**Definition 1.** *For a nonempty set $C \in \mathbb{R}^m$, the set of all minimal points in $C$ w.r.t. the ordering cone $P$ is defined as*

$$\text{Min } C := \{l \in C : C \cap (l - P) = \{l\}\}.$$

*The weakly minimal points of the set $C$ is*

$$\text{WMin } C := \{l \in C : C \cap (l - \text{int}P) = \emptyset\}.$$

Figure 1 gives a 2-dimensional example to show the difference between these two definitions of minimality.

In the MOP, for the given objective function $g(z) : \mathbb{R}^n \to \mathbb{R}^m$ $(m, n \in \mathbb{N}, m \geq 2)$, where $z \in \mathcal{Z}$, we denote by $\text{Min } g(z)$ the set of all the minimal points of the function $g$. We also call it as the Pareto frontier or Pareto-optimal set. Thus, the corresponding efficient solution or Pareto-optimal solution of $g(z)$ can be defined as

$$\text{Eff } (g(z)) := \{z \in \mathcal{Z} : g(z) \in \underset{z \in \mathcal{Z}}{\text{Min }} g(z)\}.$$

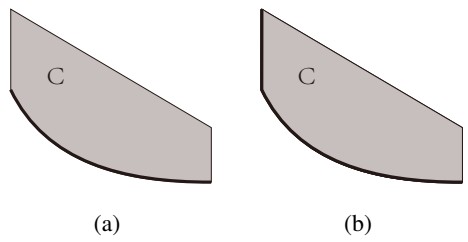

(a)                          (b)

Figure 1: The thick line in (a) indicates minimal points of C. The thick line in (b) indicates weakly minimal points of C.

Similarly, we denote by $\text{WMin } g(z)$ the set of weakly minimal points of the function $g(z)$ and by $\text{WEff } (g(z))$ the corresponding weakly efficient solution set.

**Definition 2.** *The function $g(z) : \mathbb{R}^n \to \mathbb{R}^m$ is a P-convex function if for every $z_1, z_2 \in \mathbb{R}^n$ and for every $\lambda \in [0, 1]$, we have*

$$g(\lambda z_1 + (1 - \lambda)z_2) \leq \lambda g(z_1) + (1 - \lambda)g(z_2).$$

*$g(z)$ is a strictly P-convex function, if for every $z_1, z_2 \in \mathbb{R}^n$, $z_1 \neq z_2$ and for every $\lambda \in (0, 1)$,*

$$g_i(\lambda z_1 + (1 - \lambda)z_2) < \lambda g_i(z_1) + (1 - \lambda)g_i(z_2).$$

**Remark 1.** *For a given vector-valued function $g(z)$, we have $\text{Min } g(z) \subseteq \text{WMin } g(z)$. If $g$ is strictly P-convex, we have $\text{Min } g(z) = \text{WMin } g(z)$ and $\text{WEff } (g(z)) = \text{Eff } (g(z))$.*

**Definition 3.** *Consider $\{A_n\}$ as a sequence of subsets of an Euclidean space. The set $\text{Li } A_n$ is defined as the lower limit of the sequence of sets $\{A_n\}$, that is,*

$$\text{Li } A_n := \{a \in A : a = \lim_{n \to +\infty} a_n, a_n \in A_n, \text{ for sufficiently large } n\}.$$

*The set Ls $A_n$ is defined as the upper limit of the sequence of sets $\{A_n\}$, that is,*

$$\text{Ls } A_n := \{a \in A : a = \lim_{n \to +\infty} a_n, a_n \in A_{n_k}, \text{ for } n_k \text{ as a selection of the integers.}\}.$$

*A sequence $\{A_n\}$ converges in the Kuratowski sense to the set A, when*

$$\text{Ls } A_n \subseteq A \subseteq \text{Li } A_n,$$

*and we denote this convergence by $A_n \to A$.*

### A.2 Gradient-based Optimization Algorithm

To solve an unconstrained multi-objective optimization problem, we adopt the Multiple Gradient Descent Algorithm (MGDA) [7]. MGDA finds the minimum-norm point in the convex hull composed by the gradients of multiple objectives. Specifically, MGDA performs the following two steps alternately:

**Step 1.** Compute the gradients $\nabla_z g_i(z)$ for $i = 1, \ldots, m$, and solve the following quadratic programming problem

$$\min_{\boldsymbol{\gamma}} \left\| \sum_{i=1}^{m} \gamma_i \nabla_z g_i(z) \right\|_2^2 \quad \text{s.t. } \gamma_i \geq 0, \ \sum_{i=1}^{m} \gamma_i = 1, \tag{10}$$

to determine the weights $\gamma_i$ in the current iteration. $\|\cdot\|_2$ denotes the $\ell_2$ norm of a vector and $\gamma_i$ can be viewed as a weight for the $i$th objective. To solve the QP problem (10), we can use the Frank-Wolfe algorithm [51]. Then, the gradient direction searched is computed as $d = \sum_{i=1}^{m} \gamma_i \nabla_z g_i(z)$.

**Step 2.** If $d = 0$, the MGDA stops. Otherwise, the line step is determined as the largest positive number $\nu$, with which all objectives are decreasing. Then we update $z$ as $z - \nu d$ and go to Step 1.

**Remark 2.** *The original MGDA searches the step size. However, this will result in significant computational complexity for models with many parameters such as DNN. Therefore, in Algorithm 1, similar to [51, 38], we use a fixed small step size in MGDA to reduce the computational complexity.*

## B Proofs of Theorems in Section 4

For the sake of clarity, we firstly introduce some notations from [36].

The sublevel set of the function $g(z) : \mathbb{R}^n \to \mathbb{R}^m$ at height $h \in \mathbb{R}^m$ is defined as

$$g^h := \{z \in \mathbb{R}^n : g(z) \leq h\}.$$

If $A$ is a closed convex set, then the recession cone of A is defined as

$$0^+(A) := \{d \in \mathbb{R}^n : a + td \in A, \forall a \in A, \forall t \geq 0\}.$$

The recession cone of the sublevel set of the function $g(z)$ is denoted by $H_g$.

To prove theorems in Section 4, we first prove the following theorems based on the stability analysis of MOPs [35].

**Theorem 3.** $\mathcal{Z}$ *is a nonempty closed, convex set in $\mathbb{R}^n$ and $g(z) : \mathbb{R}^n \to \mathbb{R}^m$ is a vector-valued function with $z \in \mathcal{Z}$. Then, if $g_n(z) \to g(z)$ w.r.t. the continuous convergence, we have*

$$\text{LsWMin } g_n(z) \subseteq \text{WMin } g(z).$$

*Proof.* For $l \in \text{LsWMin } g_n(z)$, there exists a subsequence $\{l_k\}$ in $\text{WMin } g_{n_k}(z)$ such that $l_k \to l$.

We assume that $l \notin \text{WMin } g(z)$. Then there exists $z \in \mathcal{Z}$ such that $g_i(z) < l_i$. Since $g_n$ continuously converges to $g$, for a sequence $\{z_k\}$ in $\mathcal{Z}$ satisfying $z_k \to z$, we have $g_{n_k}(z_k) \to g(z)$. Thus, for a sufficiently large $n$, $g_{n_k}(x_k) < l_k$. This shows a contradiction with the fact that $l_k \in \text{WMin } g_{n_k}(z)$. So $l \in \text{WMin } g(z)$ and we reach the conclusion. $\square$

**Theorem 4.** $\mathcal{Z}$ *is a nonempty closed, convex set in $\mathbb{R}^n$ and $z \in \mathcal{Z}$, $g_n(z) \to g(z)$ w.r.t. the continuous convergence. Then if $g_n(z)$ and $g(z)$ are both P-convex functions and $0^+(\mathcal{Z}) \cap H_g = \{0\}$, we have*

$$\text{Min } g(z) \subseteq \text{LiMin } g_n(z).$$

*Proof.* This result can be directly obtained from Theorems 3.1 and 3.2 of [35]. $\square$

## B.1 Proof of Theorem 1

*Proof.* To show that $F(\omega^*(\alpha), \alpha)$ is continuous on $\alpha$, we need to prove that for any convergent sequence $\alpha_n \to \bar{\alpha}$, $F(\omega^*(\alpha_n), \alpha_n)$ converges to $F(\omega^*(\bar{\alpha}), \bar{\alpha})$.

Suppose that $\{\alpha_n\}$ is a sequence in $\mathcal{A}$ satisfying $\alpha_n \to \bar{\alpha}$. Since $\arg\min_\omega f(\omega, \alpha)$ is a singleton, we have $\omega^*(\alpha_n) = \arg\min_\omega f(\omega, \alpha_n)$.

Since $\{\omega^*(\alpha)\}$ is bounded for $\alpha \in \mathcal{A}$, according to Bolzano-Weierstrass theorem, there exists a convergent subsequence $\{\omega^*(\alpha_{kn})\}$ such that $\omega^*(\alpha_{kn}) \to \bar{\omega}$ for some $\bar{\omega} \in \mathbb{R}^p$. Since $\alpha_{kn} \to \bar{\alpha}$, $f(\omega, \alpha)$ is jointly continuous, we have $\forall \omega \in \mathbb{R}^p$, $f(\bar{\omega}, \bar{\alpha}) = \lim_n f(\omega^*(\alpha_{kn}), \alpha_{kn}) \leq \lim_n f(\omega(\alpha_{kn}), \alpha_{kn}) = f(\omega(\bar{\alpha}), \bar{\alpha})$. Therefore, we obtain $\omega^*(\bar{\alpha}) = \bar{\omega}$. This means $\{\omega^*(\alpha_{kn})\}$ has only one cluster point $\omega^*(\bar{\alpha})$. Thus, $\omega^*(\alpha_n)$ converges to $\omega^*(\bar{\alpha})$ as $\alpha_n \to \bar{\alpha}$. Because $F$ is jointly continuous, we have $F(\omega^*(\alpha_n), \alpha_n) \to F(\omega^*(\bar{\alpha}), \bar{\alpha})$ as $\alpha_n \to \bar{\alpha}$. $\square$

## B.2 Proof of Theorem 2

*Proof.* To prove the first claim of Theorem 2, we firstly show that $\varphi_K(\alpha)$ continuously converges to $\varphi(\alpha)$. Suppose there exists a sequence $\{\alpha_n\}$ in $\mathcal{A}$ satisfying $\alpha_n \to \alpha$. Then for any $\varphi_K(\alpha)$ and sequence $\alpha_n$, we have

$$
\begin{aligned}
\|\varphi_K(\alpha_n) - \varphi(\alpha)\| =& \|F(\omega_K(\alpha_n), \alpha_n) - F(\omega^*(\alpha), \alpha)\| \\
\leq& \|F(\omega_K(\alpha_n), \alpha_n) - F(\omega^*(\alpha_n), \alpha_n)\| + \|F(\omega^*(\alpha_n), \alpha_n) - F(\omega^*(\alpha), \alpha)\|
\end{aligned}
$$

According to the continuity property in Theorem 1, we have $F(\omega^*(\alpha_n), \alpha_n) \to F(\omega^*(\alpha), \alpha)$ as $\alpha_n \to \alpha$. Furthermore, because $F(\cdot, \alpha)$ is uniformly Lipschitz continuous, we have

$$
\begin{aligned}
\|\varphi_K(\alpha_n) - \varphi(\alpha_n)\| =& \|F(\omega_K(\alpha_n), \alpha_n) - F(\omega^*(\alpha_n), \alpha_n)\| \\
\leq& L\|\omega_K(\alpha_n) - \omega^*(\alpha_n)\|.
\end{aligned}
$$

According to assumption $(ii)$ in Theorem 2, $\omega_K(\alpha)$ converges to $\omega^*(\alpha)$ uniformly as $K \to +\infty$. Therefore, $\varphi_K(\alpha)$ continuously converges to $\varphi(\alpha)$.

Since $\text{Min } \varphi(\alpha) \subseteq \text{WMin } \varphi(\alpha)$ and Theorem 3, we have the following set relations as

$$
\text{LsMin } \varphi_K(\alpha) \subseteq \text{LsWMin } \varphi_K(\alpha) \subseteq \text{WMin } \varphi(\alpha). \tag{11}
$$

Because $\mathcal{A}$ is a compact convex set in $\mathbb{R}^n$, $0^+(\mathcal{A}) = \{0\}$. Then, the condition $0^+(\mathcal{A}) \cap H_\varphi = \{0\}$ is naturally satisfied for function $\varphi(\alpha)$. According to assumption $(iv)$ in Theorem 2, $\varphi(\alpha)$ and $\varphi_K(\alpha)$ are both P-convex functions. Then we obtain the lower part of the set convergence from Theorem 4 as

$$
\text{Min } \varphi(\alpha) \subseteq \text{LiMin } \varphi_K(\alpha) \subseteq \text{LiWMin } \varphi_K(\alpha). \tag{12}
$$

Because $\varphi(\alpha)$ is strictly P-convex, we have $\text{WMin } \varphi = \text{Min } \varphi$ and then we get $\text{Min } \varphi_K(\alpha) \to \text{Min } \varphi(\alpha)$ according to Definition 3.

For the second claim, let $\alpha_n \in \text{Eff } \varphi_K(\alpha)$ and $\alpha_n \to \bar{\alpha}$. Since $\text{Min } \varphi_K(\alpha) \to \text{Min } \varphi(\alpha)$, we get $\varphi_K(\alpha_n) \to \varphi(\bar{\alpha})$ and $\bar{\alpha} \in \text{Min } \varphi(\alpha)$, which implies $\text{LsEff } \varphi_K(\alpha) \subseteq \text{Eff } \varphi(\alpha)$.

For the lower limit, by defining $\bar{\alpha} \in \text{Eff } \varphi(\alpha)$, the corresponding minimal point satisfies $\bar{l} = \varphi(\bar{\alpha}) \in \text{Min } \varphi(\alpha)$. Based on this theorem's first claim, there is a sequence $\{l_K\}$ in $\text{Min } \varphi_K(\alpha)$ such that $l_K \to \bar{l}$. Then we can take a bounded sequence $\{\alpha_K\}$, where $\alpha_K = \varphi_K^{-1}(l_K)$ and the subsequence of $\{\alpha_K\}$ has a cluster point. Because $\varphi(\alpha)$ is strictly P-convex, this cluster point is $\bar{\alpha}$. Then, we have $\alpha_K \to \bar{\alpha}$, which implies $\text{Eff } \varphi(\alpha) \subseteq \text{LiEff } \varphi_K(\alpha)$. Combined with the upper limit convergence, we can get $\text{Eff } \varphi_K(\alpha) \to \text{Eff } \varphi(\alpha)$. $\square$

**Remark 3.** *In fact, if we consider the weakly minimal points under the same assumptions in Theorem 1 and 2, we can still obtain similar convergence results to those in Theorem 2, i.e.,*

$$
\text{WMin } \varphi_K(\alpha) \to \text{WMin } \varphi(\alpha), \ \text{WEff } \varphi_K(\alpha) \to \text{WEff } \varphi(\alpha).
$$

*Since $\varphi(\alpha)$ is strictly P-convex, the first claim can be directly obtained from the set relations in Eqs. (11) and (12). Then, the proof of the convergence of the weakly efficient solution follows that of Theorem 2.*

# C  Few-Shot Learning

## C.1  Detailed Formulation

MAML [12], ProtoNet [55], and BOIL [40] are three representative FSL methods. We now introduce the details about how to adapt our MOML framework into them. Specifically, we will present the definition of model parameters $\alpha$, meta parameters $\omega$, and the classification loss $\mathcal{L}_F(\omega, \alpha, \mathcal{D})$ of problem (5) based on these three different algorithms.

MAML is an optimization-based meta-learning algorithm. In MAML, $\alpha$ represents the meta-initialized parameter and $\omega$ represents the task-specific parameter. In this paper, we focus on classification tasks and so the loss function $\mathcal{L}_F(\omega, \alpha, \mathcal{D})$ is the cross-entropy loss with $\alpha$ and $\omega$ on a dataset $\mathcal{D}$. Given the meta-initialized parameter $\alpha$ of the backbone, we use $\alpha$ to initialize task-specific parameters $\omega_0$ and update $\omega_0$ to $\omega_K$ in $K$ steps on the support set for the LL subproblem. Then, we compute the loss on the query set in the UL subproblem by using $\omega_K$ for the corresponding task. Thus, we can update $\alpha$ and find universally good meta-initialized parameters that can quickly adapt to new tasks with a small number of samples. In problem (5), we find the meta-initialized parameters $\alpha$ that not only have good performance but also be robust to adversarial attacks when adapting to new tasks with a few examples.

ProtoNet is a metric-based FSL algorithm. In ProtoNet, $\alpha$ represents the parameter of the embedding function $f(x; \alpha)$, which encodes inputs into a vector space $\mathcal{V}$. $\omega^k$ represents the prototype of the $k$th class, which can be considered as a class center. Suppose there are $n$ labeled examples in dataset $\mathcal{D}$ and $C$ is the number of classes. Then, given a distance function $d$ in $\mathcal{V}$, the distribution over classes for one input $x$ is a softmax over the inverse of distances to the prototypes in vector space $\mathcal{V}$,

$$P(y = k \mid x; \alpha, \omega) = \frac{\exp(-d(f(x; \alpha), \omega^k)}{\sum_{k'=1}^{C} \exp(-d(f(x; \alpha), \omega^{k'}))},$$

where $\omega$ represents the set of all prototypes. Suppose $\mathcal{D}_k \subset \mathcal{D}$ is the set of examples labeled with class $k$ and $n_k$ is the corresponding number of examples. Then, the loss function is the negative log-likelihood

$$\mathcal{L}_F(\omega, \alpha, \mathcal{D}) = -\frac{1}{n} \sum_{k=1}^{C} \sum_{x_i \in \mathcal{D}_k} \log P(y_i = k \mid x_i; \alpha, \omega),$$

where $k$ is the true class. In the LL subproblem, by minimizing this loss function *w.r.t.* $\omega$, we have a close form solution $\omega^k = \frac{1}{n_k} \sum_{x_i \in \mathcal{D}_k} f(x_i; \alpha)$. In the UL subproblem, we minimize this loss *w.r.t.* $\alpha$ on the query set.

BOIL is also an optimization-based meta-learning algorithm. Similar to MAML, BOIL also aims to find universally good meta-initialized parameters. However, BOIL updates only the feature extractor of the model and freezes the classification layer in the LL subproblem. In the UL subproblem, BOIL updates the meta-initialized parameters of the feature extractor and classification layer, which is the same as MAML. Specifically, consider task-specific parameters $\omega = \{\theta, \psi\}$, where $\theta$ denotes the parameter of body of the model and $\psi$ represents the classifier parameter, respectively. $\alpha = \{\alpha_\theta, \alpha_\psi\}$ represent the meta-initialized parameters. In BOIL, we update $\alpha$ in the UL subproblem and only update $\theta$ in LL subproblem.

## C.2  Experimental Setting

Experiments are conducted on two FSL benchmark datasets, CUB-200-2011 (referred to as CUB) [62] and *mini*-ImageNet [61]. The CUB dataset contains 200 classes and 11,788 images in total. Following [18], we randomly split this dataset into a base dataset containing 100 classes, a validation dataset containing another 50 classes, and a novel dataset containing the rest 50 classes. The *mini*-ImageNet dataset contains 100 classes with 600 images per class, sampled from the ImageNet dataset [6]. By following [46], this dataset is partitioned into 64, 16, and 20 classes for the base, validation, and novel datasets, respectively.

For all methods, each task is a 5-way $k$-shot classification problem, where $k$ equals 1 or 5. The input images are resized to $84 \times 84$ for both two datasets and applied data augmentation including random crop, random horizontal flip, and color jitter. A four-layer convolutional neural network (Conv-4) is

used as the backbone, which consists of four blocks each of which consists of a convolution layer with 64 kernels of size $3 \times 3$, stride 1, and zero padding, a batch normalization layer, a ReLU activation function, and a max-pooling layer with the pooling size $2 \times 2$. After the backbone, a fully-connected linear layer with 5 neurons is used as a classifier to output the prediction for the input image. The Adam optimizer [24] with the learning rate 0.001 is used for the optimization.

In the meta training process, we randomly sample $k$ and 16 instances per class as the support set and the query set, respectively, in each episode. The adversarial attack on the query set is performed by the PGD attack with a perturbation size $\epsilon = 2/255$ and it takes 7 iterative steps with the step size of $2.5\epsilon$. In the meta testing process, we generate 600 5-way $k$-shot tasks from $\mathcal{D}_{novel}$, where each task has $k$ samples for the training and 16 samples for testing. We compute the average results on all the 600 testing tasks. We compare with the vanilla FSL models (*i.e.*, MAML, ProtoNet, and BOIL) since problem (5) can reduce to each of them when there is only the first objective in its UL subproblem. All these experiments are conducted on one single NVIDIA Tesla V100S GPU.

### C.3 Experimental Results on the CUB Dataset

The average results over 600 testing tasks on the CUB dataset are shown in Table 4. From the results, we can get similar conclusions to the experimental results on *mini*-ImageNet dataset. The SOML and the proposed MOML can significantly improve the PGD accuracy. The B-score of MOML is higher than SOML, which indicates that the porposed MOML is better than the single-objective formulation in SOML.

Table 4: Classification accuracy (abbreviated as "Clean Acc.") and PGD accuracy (abbreviated as "PGD Acc.") on the CUB dataset for 5-way $k$-shot FSL. The best result in each group of methods is highlighted in **bold** and the best result in each setting is annotated with underline.

| Setting | Model | Clean Acc. | PGD Acc. | B-score |
|---|---|---|---|---|
| **1-shot** | MAML [12] | 54.62±0.87 | 3.92±0.49 | 7.28±0.61 |
| | MAML+SOML | 49.60±0.81 | 36.42±0.87 | 41.89±0.84 |
| | MAML+MOML (**ours**) | 48.66±0.87 | 38.37±0.90 | **42.75**±0.89 |
| | ProtoNet [55] | 52.93±0.91 | 2.00±0.29 | 3.53±0.47 |
| | ProtoNet+SOML | 48.04±0.91 | 28.53±0.85 | 35.42±0.90 |
| | ProtoNet+MOML (**ours**) | 42.26±0.89 | 32.19±0.82 | **36.24**±0.85 |
| | BOIL [40] | 61.79±0.94 | 6.53±0.48 | 11.81±0.61 |
| | BOIL+SOML | 54.29±0.83 | 33.65±0.67 | 41.34±0.71 |
| | BOIL+MOML (**ours**) | 52.15±0.93 | 40.44±0.94 | **45.55**±0.94 |
| **5-shot** | MAML [12] | 75.57±0.72 | 8.76±0.78 | 15.61±0.76 |
| | MAML+SOML | 68.50±0.69 | 52.96±0.87 | 59.63±0.77 |
| | MAML+MOML (**ours**) | 67.57±0.78 | 55.26±0.87 | **60.68**±0.83 |
| | ProtoNet [55] | 78.01±0.71 | 1.89±0.21 | 3.58±0.56 |
| | ProtoNet+SOML | 72.51±0.68 | 52.61±0.77 | 60.81±0.72 |
| | ProtoNet+MOML (**ours**) | 71.10±0.74 | 56.11±0.87 | **62.73**±0.76 |
| | BOIL [40] | 78.97±0.67 | 14.12±0.57 | 23.75±0.60 |
| | BOIL+SOML | 76.25±0.60 | 44.86±0.81 | 56.28±0.73 |
| | BOIL+MOML (**ours**) | 71.03±0.74 | 56.05±0.84 | **62.65**±0.81 |

## D  Semi-Supervised Domain Adaptation

### D.1  Detailed Formulation of Problem (6)

We give detailed formulations of problem (6) when adapting to MME [49] and DSAN [74]. Let $\omega = \{\theta, \psi\}$, where $\theta$ denotes the parameter of backbone and $\psi$ represents the classifier parameter. Correspondingly, let $\alpha = \{\alpha_\theta, \alpha_\psi\}$, where $\alpha_\theta$ and $\alpha_\psi$ represent the initialization of $\theta$ and $\psi$, respectively. Assume $f(\theta, \alpha_\theta, x)$ denotes the feature representation of the input $x$ extracted by the backbone $\theta$ with the initialization $\alpha_\theta$. $C$ denotes the number of classes. $n_{\mathcal{S}}$ and $n_{\mathcal{T}}^u$ denote the size of $\mathcal{D}_{\mathcal{S}}$ and $\mathcal{D}_{\mathcal{T}}^u$, respectively.

When adapting problem (6) to different domain adaptation methods, the biggest difference is the design of the alignment loss $\mathcal{L}_A(\omega, \alpha, \mathcal{D}_S, \mathcal{D}_T^u)$. Combining with MME [49], we can define the alignment loss $\mathcal{L}_A$ as the entropy loss to find domain-invariant prototypes

$$\mathcal{L}_A(\omega, \alpha, \mathcal{D}_T^u) = \sum_{i=1}^{n_T^u} \sum_{c=1}^{C} p(y = c \mid x_i^T) \log p(y = c \mid x_i^T), \tag{13}$$

where $\sigma(\cdot)$ denotes the softmax function and $p(y = c | x) = \left[ \sigma\left( \frac{\psi^T f(\theta, \alpha_\theta, x)}{\| f(\theta, \alpha_\theta, x) \|} \right) \right]_c$ computes the conditional probability that $x$ belongs to class $c$. Obviously, $\psi$ can be considered as the prototypes based on the cosine distance. Although Eq. (13) is not explicitly dependent on source domain data $\mathcal{D}_S$, the prototypes $\psi$ are computed on $\mathcal{D}_S$ in previous iterations. Therefore, minimizing Eq. (13) (*i.e.*, maximizing the entropy) can promote the prediction distribution of samples from unlabeled target domain to uniform distribution, *i.e.*, all target features are close to the prototypes, which indicates learning the domain-invariant prototypes.

When adapting to DSAN [74], the alignment loss $\mathcal{L}_A$ is defined as the local maximum mean discrepancy to measure the difference between two domains. Thus, following DSAN, $\mathcal{L}_A$ is formulated as

$$\mathcal{L}_A(\theta, \alpha_\theta, \mathcal{D}_S, \mathcal{D}_T^u) = \frac{1}{C} \sum_{c=1}^{C} \Big[ \sum_{i=1}^{n_S} \sum_{j=1}^{n_S} w_{i,c}^S w_{j,c}^S \, k(f(\theta, \alpha_\theta, x_i^S), f(\theta, \alpha_\theta, x_j^S))$$
$$+ \sum_{i=1}^{n_T^u} \sum_{j=1}^{n_T^u} w_{i,c}^T w_{j,c}^T \, k(f(\theta, \alpha_\theta, x_i^T), f(\theta, \alpha_\theta, x_j^T))$$
$$- 2 \times \sum_{i=1}^{n_S} \sum_{j=1}^{n_T^u} w_{i,c}^S w_{j,c}^T \, k(f(\theta, \alpha_\theta, x_i^S), f(\theta, \alpha_\theta, x_j^T)) \Big],$$

where $k(\cdot, \cdot)$ denotes the kernel function. Here $w_{i,c}^S = \frac{y_{i,c}}{\sum_{j=1}^{n_S} y_{j,c}}$ represents the possibility of sample $x_i^S$ belonging to class $c$, where $y_{i,c}$ is the $c$-th element of the one-hot label vector of $x_i^S$. The definition of weight $w_{i,c}^T$ is similar, while we use the prediction distribution as the pseudo label of $x_i^T$ from $\mathcal{D}_T^u$ since its true label is unavailable. Specifically, when combining with DSAN, problem (6) is formulated as

$$\min_{\alpha_\theta, \psi} \; (\mathcal{L}_D(\{\theta^*(\alpha_\theta), \psi\}, \mathcal{D}_S), \mathcal{L}_D(\{\theta^*(\alpha_\theta), \psi\}, \mathcal{D}_T^l), \mathcal{L}_A(\theta, \alpha_\theta, \mathcal{D}_S, \mathcal{D}_T^u))$$
$$\text{s.t. } \theta^*(\alpha_\theta) = \arg\min_\theta \mathcal{L}_A(\theta, \alpha_\theta, \mathcal{D}_S, \mathcal{D}_T^u).$$

### D.2  More Details of Baselines

We compare the proposed MOML with eight state-of-the-art baselines, including one Semi-Supervised Learning (SSL) method (*i.e.*, ENT), four Unsupervised Domain Adaptation (UDA) methods (*i.e.*, DANN, ADR, CDAN, DSAN) and three Semi-Supervised Domain Adaptation (SSDA) methods (*i.e.*, MME, Meta-MME, APE). **ENT** [16] is a SSL method trained on labeled source data and unlabeled target data by minimizing the classification loss and the entropy of the predictive distribution. **DANN** [14] uses a domain classifier such that domains can not be discriminated from each other by adversarial training. **ADR** [50] learns discriminative features by multiple classifiers with different dropout rates in an adversarial training manner. **CDAN** [32] tries to align source and target domains in the feature space conditioned on the output of the classifier via adversarial training. Instead of adversarial learning, **DSAN** [74] aims to align the same category of different domains via the local maximum mean discrepancy. For these four UDA methods, we appropriately modify them so that they can be trained with the labeled target domain. **MME** [49] aims to learn domain-invariant prototypes by maximizing the entropy on unlabeled target data and minimizing the entropy on labeled source and target data. **Meta-MME** [27] reformulates the MME method as a bi-level optimization problem, while its formulation is significantly different with the proposed MOML (*i.e.*, problem (6)). **APE** [23] aims to minimize the intra-domain discrepancy within the target domain to improve the domain alignment.

### D.3 Experimental Setting

Experiments are conducted on the Office-31 dataset [60]. This dataset[1] consists of three domains: Amazon, DSLR, and Webcam, abbreviated as **A**, **D**, and **W**, respectively. It contains 4,110 labeled images in total and each domain consists of 31 categories.

We use the ResNet-50 model [17] pretrained on the ImageNet dataset as the backbone network followed by a Fully-Connected (FC) layer. The same network architecture is used for all baseline methods. All baselines use the same experimental setting as the original methods. For the training of all the DSAN-related models, the SGD optimizer with the learning rate $10^{-3}$, the momentum 0.9 and the weight decay $5 \times 10^{-4}$ is used for optimization. The batch size is set to 96, including 32 images in the source, labeled target, and unlabeled target domains, respectively. For all the MME-related models, we implement them based on the public code base[3] and use the same experimental settings as the original MME method. All the experiments are conducted on one single NVIDIA Tesla V100S GPU.

## E  Multi-Task Learning

### E.1  Datasets

Experiments are conducted on the NYUv2 [53], Office-31 and Office-Home [60] datasets. The NYUv2 dataset is an indoor scene RGB-D image dataset, which consists of three tasks: 13-class semantic segmentation, depth estimation, and surface normal prediction. We use the NYUv2 dataset pre-processed by [31]. It contains 795 and 654 labeled images for training and test, respectively, and the size of each image is $288 \times 384$. The Office-31 dataset has been introduced in Appendix D.3 and we consider the classification problem on each domain as a task. Thus, there are three tasks on the Office-31 dataset. The Office-Home dataset[4] contains four tasks: artistic images, clip art, product images, and real-world images, abbreviated as **Ar**, **Cl**, **Pr**, and **Rw**, respectively. This dataset has 15,500 labeled images in total and each task consists of 65 object categories in office and home settings.

### E.2  Experimental Settings

**NYUv2**  The ResNet-50 pretrained on the ImageNet dataset is used as the backbone to extract features and $m$ Atrous Spatial Pyramid Pooling (ASPP) [3] modules are used as the decoder for task-specific outputs. For the DMTL architecture, the multi-task learning model adopts the widely used hard-sharing or equivalently the multi-head structure. For the MTAN architecture, we add $m$ task-specific attention networks into the backbone based on the DMTL architecture. We implement both DMTL and MTAN architectures based on the public code base[5]. The MTL model with parameter $\omega$ in problem (7) is trained by the Adam optimizer [24] with the learning rate as $10^{-4}$ and weight decay as $10^{-5}$. The loss weight $\boldsymbol{\alpha}$ in problem (7) is initialized as $(\frac{1}{3}, \frac{1}{3}, \frac{1}{3})^T$ and is optimized by the Adam optimizer with the learning rate as $10^{-4}$. For SOML and the proposed MOML method, we randomly split 795 training images into two parts: 636 for training and the rest 159 for validation and test on the same test dataset as all baselines. A batch size 8 is used for DMTL architecture and 4 for MTAN architecture. All the experiments are conducted on one single NVIDIA Tesla V100S GPU.

**Office-31 and Office-Home**  The experimental settings of Office-31 and Office-Home datasets are similar to those of the NYUv2 dataset. The ResNet-18 pretrained on the ImageNet dataset is used as the backbone to extract features and $m$ linear layers are used for task-specific classification. We also add $m$ task-specific attention networks to build the MTAN architecture. The MTL model is trained by the Adam optimizer [24] with the learning rate as $10^{-4}$. The loss weight $\boldsymbol{\alpha}$ is initialized with equal values and optimized by the Adam optimizer with the learning rate as $10^{-3}$. Both the Office-Home and Office-31 datasets are split into three parts, including 60% for training, 20% for validation, and the remaining 20% for testing. All the baselines are trained on training and validation datasets. We set

---

[1]https://www.cc.gatech.edu/~jhoffman/domainadapt/#datasets_code
[3]https://github.com/VisionLearningGroup/SSDA_MME
[4]https://www.hemanthdv.org/officeHomeDataset.html
[5]https://github.com/lorenmt/mtan/tree/master/im2im_pred/model_resnet_mtan

the batch size to 64 for both Office-31 and Office-Home datasets. All the experiments are conducted on one single NVIDIA Tesla V100S GPU.

### E.3 Experimental Results on the Office-31 and Office-Home Datasets

Experimental results on the Office-31 and Office-Home datasets are shown in Table 5. Firstly, we notice that SOML achieves comparable performance with state-of-the-art baselines under both DMTL and MTAN architectures on both Office-31 and Office-Home datasets, which means the proposed bi-level formulation (*i.e.*, problem (7) with weighted combined objectives in the UL subproblem) is competitive when comparing with the baselines with single-objective formulation. Secondly, the proposed MOML slightly outperforms SOML in all cases. It indicates the multi-objective formulation in the UL subproblem is better than the single-objective formlation in SOML. Finally, compared with state-of-the-art baselines, the proposed MOML achieves the best result in some tasks, such as the best classification accuracy 88.20% in task **A** on the Office-31 dataset.

Table 5: Classification accuracy (%) on the Office-31 and Office-Home datasets. The best combinations of the architecture and weighting strategy are highlighted in **bold**. The best results for each task on each measure are annotated with underlines.

| Architecture | Weighting Strategy | Office-31 | | | | Office-Home | | | | |
|---|---|---|---|---|---|---|---|---|---|---|
| | | **A** | **D** | **W** | **Avg** | **Ar** | **Cl** | **Pr** | **Rw** | **Avg** |
| DMTL | EW | 87.17 | 98.36 | **99.44** | 94.99 | 68.88 | 80.93 | **91.73** | 81.72 | 80.81 |
| | DWA [31] | 87.52 | 99.18 | **99.44** | 95.38 | 70.39 | 79.95 | 90.36 | **82.05** | 80.69 |
| | MGDA [51] | 87.52 | 99.18 | **99.44** | 95.38 | 69.44 | 79.30 | 91.63 | 81.72 | 80.52 |
| | PCGrad [71] | 87.00 | 98.36 | 98.33 | 94.56 | 68.31 | 80.71 | 90.57 | 81.94 | 80.38 |
| | SOML | 87.35 | **100** | 98.89 | 95.41 | **70.77** | 81.14 | 90.46 | 80.97 | 80.84 |
| | MOML (**ours**) | **87.69** | 99.18 | **99.44** | **95.43** | 69.63 | **81.79** | 91.20 | **82.05** | **81.17** |
| MTAN [31] | EW | 87.52 | 98.36 | **99.44** | 95.10 | 69.63 | 80.60 | 91.94 | 82.91 | 81.27 |
| | DWA [31] | 87.52 | **100** | **99.44** | 95.65 | 69.63 | 81.14 | 91.10 | 82.48 | 81.09 |
| | MGDA [51] | 87.35 | 99.18 | **99.44** | 95.32 | 69.25 | **81.36** | 91.73 | 82.81 | 81.29 |
| | PCGrad [71] | 87.69 | **100** | **99.44** | 95.71 | 69.25 | **81.36** | **92.37** | 82.27 | 81.31 |
| | SOML | 87.52 | **100** | **99.44** | 95.65 | 70.77 | 81.25 | 91.20 | 82.59 | 81.45 |
| | MOML (**ours**) | **88.20** | **100** | **99.44** | **95.88** | **70.96** | 81.14 | 92.05 | **83.02** | **81.80** |

# F  Neural Architecture Search

## F.1  Experimental Settings

The search space and training procedure of MOML adopt the same settings as DARTS [29]. Specifically, in both normal and reduction cells, the set of operations $\mathcal{O}$ contains eight operations, including $3 \times 3$ separable convolutions, $5 \times 5$ separable convolutions, $3 \times 3$ dilated separable convolutions, $5 \times 5$ dilated separable convolutions, $3 \times 3$ max pooling, $3 \times 3$ average pooling, identity, and zero. Half of the training set is used for training a model, and the other half is for the validation. A small network of 8 cells is trained with the batch size as 64 and 16 initial channels for 50 epochs. The Adam optimizer [24] with the learning rate $3 \times 10^{-4}$, the momentum $\beta = (0.5, 0.999)$, and the weight decay $1 \times 10^{-3}$ is used to update $\alpha$ in the UL subproblem. The SGD optimizer with the decayed learning rate down from $0.025$ to $0$ by a cosine schedule, the momentum $0.9$, and the weight decay $3 \times 10^{-4}$ is used to update $\omega$ in the LL subproblem.

In the evaluation stage, a neural network of 20 searched cells is trained on the full training set for 600 epochs with the batch size as 96, the initial number of channels as 36, the length of a cutout as 16, the dropout probability as 0.2, and auxiliary towers of weight as 0.4. The full testing set is used for testing. Adversarial examples are generated using the PGD attack with the perturbation size $\epsilon = 1/255$ and the PGD attack takes 10 iterative steps with the step size of $2.5\epsilon$ as suggested in [26]. All the experiments are conducted on one single NVIDIA Tesla V100S GPU.

Table 6: Comparison between MOML and DARTS and other three state-of-the-art baselines on the CIFAR-10 dataset. ↑ indicates that a larger value is better, while ↓ implies that a lower value is better. "{DARTS-C#channels}" means that the architecture searched by DARTS is evaluated with the initial number of channels as "channels". "{SOML-V#size-C#channels}" means that the architecture searched by SOML with $L$ as "size" is evaluated by the initial number of channels as "channels". "{MOML-V#size}" denotes the architecture searched by MOML with $L$ as "size". The B-score, which is defined in Section 5.1, measures the balance between the clean accuracy and PGD accuracy when the numbers of parameters in different architecture are comparable.

| Architecture | Params (MB) ↓ | Clean Acc. (%) ↑ | PGD Acc. (%) ↑ | B-score |
|---|---|---|---|---|
| SNAS (moderate) [67] | 2.8 | 97.15 | - | - |
| RC-DARTS-C42 [21] | 3.3 | 97.19 | - | - |
| ENAS [42] | 4.6 | 97.11 | - | - |
| DARTS-C26 | 1.787 | 96.91 | 28.45 | 43.98 |
| SOML-V1-C38 | 1.750 | 96.36 | 40.20 | 56.73 |
| MOML-V1 | 1.754 | 96.48 | 42.66 | **59.16** |
| DARTS-C30 | 2.354 | 97.13 | 31.53 | 47.60 |
| SOML-V2-C42 | 2.402 | 97.03 | 31.44 | 47.49 |
| MOML-V2 | 2.367 | 97.18 | 36.15 | **52.69** |
| DARTS-C34 | 2.998 | 97.34 | 30.31 | 46.22 |
| SOML-V3-C36 | 3.018 | 97.18 | 35.36 | **51.85** |
| MOML-V3 | 3.018 | 97.25 | 35.22 | 51.71 |

## F.2 Experimental Results and Analysis

Table 6 compares the proposed MOML with the DARTS method and other three state-of-the-art baselines on the CIFAR-10 dataset [25]. We search for neural networks with different expected sizes (*i.e.*, different $L$'s used in Eq. (8)) via the MOML method. To make the network size searched by DARTS comparable with that of MOML under different settings, we use different numbers of initial channels in DARTS during the evaluation process. Hence, according to the parameter size (*i.e.*, $L$=1, 2, and 3), we split the results of DARTS, SOML and MOML into three groups.

Comparing with three state-of-the-art baselines (*i.e.*, SNAS, RC-DARTS and ENAS), the proposed MOML can search an architecture with similar or less parameter size but higher clean accuracy than all baselines. Comparing with DARTS, MOML achieves a better trade-off among accuracy, network size, and robustness. With a comparable number of parameters, the MOML method improves the robustness while test accuracy are comparable or even slightly better. For example, compared MOML-V1 with DARTS-C26, the PGD accuracy increases by about 14%, while the clean test accuracy only drops around 0.5%. In addition, comparing with SOML, MOML has a higher B-score when $L$ equals 1 and 2, and has a comparable B-score when $L$ equals 3. It indicates MOML with multi-objective formulation in the UL subproblem can achieve a better trade-off between the clean accuracy and PCG accuracy than SOML with the single-objective formulation. So experimental results in Table 6 show that the MOML method can search more robust architectures with similar model size and comparable classification accuracy than the DARTS method and other state-of-the-art baselines.