# OpenReview forum: "Multi-Objective Meta Learning"
_NeurIPS.cc/2021/Conference — NeurIPS 2021 Poster_

### Official Review · Reviewer_3cBH · 2021-07-01

**Rating:** 5
**Confidence:** 4

**Summary:**

This paper considers the problem of meta-learning with multiple objectives in the optimization problem for the meta-parameters.
A novel algorithm is proposed for this problem, and the approach is evaluated on a wide set of tasks and applications.

**Limitations And Societal Impact:**

The authors should provide some discussion on the societal impacts of the work.

**Main Review:**

Strengths:
- The motivation is clear and straightforward. The problem discussed in the paper is of interest to many ML applications, as seen in the vast experiments covering FSL, MTL, NAS, and DA.
- The paper is well written, although many details are missing from the main text due to the page limit.
- The authors provide theoretical results on the convergence of the proposed Alg., which further motivates the approach.

Questions/suggestions/Weaknesses:
- The broader scope of MOO is to find a set of solutions with different trade-offs across the Pareto front. This is not discussed in this paper and the approach only generates a single solution. So what is the advantage/benefit or conceptual difference from using MTL methods in the UL optimization?
- In relation to the previous point, in sections 5.1 and 5.2, it would be good to evaluate other MTL approaches for weighting the losses (aside from linear scalarization/SOML), such as GradNorm, DWA, PCGrad, etc.
- In all experiments, it would have been beneficial to discuss or visualize the corresponding losses (in terms of dominated solutions w.r.t baselines, Pareto frontier, etc.).
- In relation to the previous point, the standard MOO metric is hyper-volume (HV). It would be useful to report the HV of multiple runs and compare it to baseline methods.
- In Tables 2 and 3, please report standard errors or some other measure of variability for the results.
- Section 5.3: At least for the SOML approach, how come the alpha corresponds to the easiest task (lowest loss) is not learned to be one while all other weights are set to zero (since the alphas are directly optimized on the validation loss)? This will minimize the weighted validation loss. Could you please explain why this is not the case?
- The improvements in all experiments are generally fairly limited compared to baselines.
- If I understand correctly, there is no validation split for baseline methods. How did you select hyper-parameters (HP) for baselines methods and for your method? Was there any optimization/search of HPs?


**Time Spent Reviewing:**

4-5 hours

---

> ### Author Response · Authors · 2021-08-11
> **Response to Reviewer 3cBH**
>
> We thank you for the detailed and constructive comments. In the following we will make a response. If you think we have not addressed your concerns, we look forward to further discussions with you.
>
> **Q1**: What is the advantage/benefit or conceptual difference from using MTL methods in the UL optimization?
>
> **A1**: Firstly, the MOML method broaden the application scope of meta learning since many real-world problems naturally have multiple objectives such as the cases discussed in the paper. Secondly, MOML generalizes the vanilla meta learning since meta learning formulated as a bi-level optimization problem is a special case of MOML when the upper-level problem has only one objective. Such generalization brings many issues, for example, how to solve it efficiently, how to analyze it in terms of, for example, the convergence, et al. Such issues are more complicated to study than the vanilla meta learning. Thirdly, formulating under the MOML can save a large computational cost. For example, under the bi-level formulation in the vanilla meta learning, to handle multiple meta-objectives, we need to tune weights among meta-objectives, which is computational demanding, while MOML has no such need.
>
> **Q2**: It is recommended to evaluate other MTL methods for weighting the losses in sections 5.1 and 5.2.
>
> **A2**: We will conducts these experiments to further show the effectiveness of the proposed MOML method in the revision.
>
> **Q3**: It is recommended to discuss or visualize the corresponding losses (in terms of dominated solutions w.r.t baselines, Pareto frontier, etc.).
>
> **A3**: We take the 5-way 1-shot and 5-way 5-shot few-shot learning setting on the mini-ImageNet dataset as examples and we find the approximate Pareto frontier via grid search on the weights of MAML+SOML. The experimental results are shown in [https://www.dropbox.com/s/2fjbjx3ybz7ifzd/NeurIPS2021.pdf?dl=0](https://www.dropbox.com/s/2fjbjx3ybz7ifzd/NeurIPS2021.pdf?dl=0). According to the results, we can see that the solution of the proposed MOML falls very well on the Pareto frontier. We will add this discussion in the revision.
>
> **Q4**: It is recommended to apply the hyper-volume as an evaluation metric for comparison.
>
> **A4**: We will leverage this hyper-volume metric to evaluate the proposed MOML method.
>
> **Q5**: It is recommended to report the standard errors in tables 2 and 3.
>
> **A5**: We will add the standard errors in tables 2 and 3 in the revision.
>
> **Q6**: In section 5.3, how come the alpha corresponds to the easiest task (lowest loss) is not learned to be one while all other weights are set to zero?
>
> **A6**: According to Eq. (7) in our submission, $\alpha$, the loss weighting in the lower-level subproblem, is optimized by the validation losses in the upper-level subproblem. Furthermore, those validation losses are minimized in a multi-objective problem instead of the sum weighted by $\alpha$. Therefore, the case you are considering will not happen.
>
> **Q7**: The improvements in all experiments are limited.
>
> **A7**: Firstly, we have compared the proposed MOML method with the state-of-the-art baselines in all cases. Secondly, to the best of our knowledge, although the improvements seem limited, these have been a large progress especially in the semi-supervised domain adaptation and multi-task learning problems.
>
> **Q8**: How to select hyper-parameters?
>
> **A8**: Actually, for all baselines in all cases, we directly use the optimal hyper-parameters in their original methods. And for our proposed MOML method, we did not search for hyper-parameters, just use the same hyper-parameters with baselines.

---

> > ### Comment · Reviewer_3cBH · 2021-08-17
> > **Response to authors rebuttal**
> >
> > I would like to thank the authors for the detailed response. After reading the reviews and the response provided by the authors, I would like to keep my score at 5.

---

### Official Review · Reviewer_CNYc · 2021-07-01

**Rating:** 4
**Confidence:** 3

**Summary:**

This work introduces a new meta-learning approach for multi-objective settings. Compared to existing works, the proposal does not require combining these multiple objectives into a single summary via a difficult-to-tune weight parameter.

**Limitations And Societal Impact:**

yes

**Main Review:**

Originality: I'm not an expert on existing approaches to meta-learning when there are multiple objectives, and so am not well qualified to assess the originality of this work.


Quality:

1.  When reading Theorem 1, I was surprised that it didn't require any sort of continuity condition on the function $f$. From looking at the proof in the supplement, it seems that there may be an issue with the current argument unless a further regularity condition is imposed in the theorem statement. In particular, the current proof claims that the accumulation point $\bar{\omega}$ is equal to $\omega^*(\bar{\alpha})$, where both of these quantities are defined within the proof.

However, a simple counterexample that satisfies all of the conditions of the currently stated Theorem 1 shows this to be false. In particular, you could have $\mathcal{A}=[0,1]$, $p=1$, and $f(\omega,\alpha)=1_{\alpha\not=\omega,\omega\in\mathbb{Q}} + 1_{\alpha\not=2\omega,\omega\not\in\mathbb{Q}}$, where $\mathbb{Q}$ is the set of rationals. Then, we can choose a set of rational numbers $\alpha_n$ converging to an irrational number $\bar{\alpha}$, and it will be the case that $\omega^*(\alpha_n)=\alpha_n\rightarrow\bar{\alpha}$, and this limit is not equal to $\omega^*(\bar{\alpha})=2\bar{\alpha}$. Granted the $f$ in this example is pathological, but that's the point -- the current conditions of the theorem don't disallow such an $f$.

2. More generally, I think more discussion is needed in the paper about the extent to which the regularity conditions assumed in the theorems are actually satisfied by the neural network parameterizations used in the experiments --- in particular, Conditions (ii) and (iv) in Theorem 2 both seem potentially problematic in neural network settings. In my view, the current presentation is somewhat misleading since it seems to imply that there are convergence guarantees for the implemented methods, which it seems there are not. E.g., it seems to me that this sentence in the abstract requires some qualification:

> Theoretically, we prove convergence properties of the proposed gradient-based optimization algorithm.

3. It's notable that, in Table 1, your main comparison seems to focus on the B-score criterion that you yourself have defined. Is there any more widely accepted score that you could use to evaluate the performance of your method?

It could also be worth finding a way to indicate the settings where XX+MOML (yours) dominates XX+SOML. It looks like this occurs for 1-shot BOIL and 5-shot BOIL. In all other settings, it looks like you outperform SOML according to PGD Acc but underperform in terms of Clean Acc. This makes me wonder if increasing the weight that SOML places on PGD Acc would lead to an approach that dominates yours in some settings (i.e., outperforms both in terms of Clean Acc and PGD Acc).


Clarity:

The presentation of the problem of interest at the beginning of Section 3 could be improved. First, it's critical to formally define the meaning of $\min g(z)$ for a vector-valued function before or immediately after Eq. 1. At present, this isn't introduced until 2 pages later (top of page 5). Second, I think it would help to briefly mention what role $\alpha$ and $\omega$ will play in settings of interest, e.g. few shot learning.


Significance:

Multi-objective settings are common in practice, and so the possibility of a meta-learning procedure that does not require selecting a difficult-to-tune weight function is appealing.

**Time Spent Reviewing:**

2.5

---

> ### Author Response · Authors · 2021-08-11
> **Response to Reviewer CNYc**
>
> We thank you for the detailed and constructive comments. In the following we will make a response. If you think we have not addressed your concerns, we look forward to further discussions with you.
>
> **Q1**: Question about Theorem 1.
>
> **A1**: We do realized that we had missed an assumption that the mapping $(\omega, \alpha) \mapsto \arg\min_\omega f(\omega,\alpha)$ should be jointly continuous. We need to use this assumption to get the result $\omega^*(\bar{\alpha}) = \bar{\omega}$ in Appendix B.1 (line 688). Then we can get $\forall \omega \in \mathbb{R}^p$, $f(\bar{\omega},\bar{\alpha}) = \lim_n f(\omega^*(\alpha_{kn}),\alpha_{kn}) \le \lim_n f(\omega(\alpha_{kn}),\alpha_{kn}) = f(\omega(\bar{\alpha}),\bar{\alpha})$. So we have $\omega^*(\bar{\alpha}) = \bar{\omega}$. We will add this assumption in Sections 3 and 4.3 in the revision.
>
> **Q2**: Question about Theorem 2.
>
> **A2**: Condition (ii) actually shows the convergence of the lower-level subproblem. For example, If $f$ is strongly convex, then many gradient-based algorithms can achieve linear convergence to $\omega^*(\alpha)$. It is a common assumption in many bi-level optimization methods such as the following references [1, 2]. The condition (iv) is unfortunately needed because it plays an important role in the set convergence analysis. In our future work, we will study whether it will converge to the Pareto stationary set if the upper-level subproblem is non-convex.
> The convergence properties we provided show the theoretical guarantee for problem (4). We will describe it more specifically in the revision.
>
> [1] Investigating Bi-Level Optimization for Learning and Vision from a Unified Perspective: A Survey and Beyond. Liu et. al.
>
> [2] A Generic First-Order Algorithmic Framework for Bi-Level Programming Beyond Lower-Level Singleton. Liu et. al. ICML, 2020.
>
> **Q3**: Is there any more widely accepted score instead of B-score to evaluate the performance?
>
> **A3**: Actually, we are unaware of other metrics to evaluate the performance on the two objectives (i.e., clean and PGD accuracy), hence we design the B-score which is inspired by the widely-used F1-score. If there is some suitable metric, we are more than happy to apply it to evaluate the performance.
>
> **Q4**: In the case of few-shot learning, if increasing the weight that SOML places on PGD Acc, would it leads to an approach that dominates the MOML method?
>
> **A4**: We think that the answer is No. To verify it, we show in the following table the experimental results of SOML when increasing the weight on the PGD objective. We use the 5-way 5-shot setting on the mini-ImageNet and CUB datasets as an example, where $\gamma$ denotes the weight of the PGD objective on the upper-level subproblem. It is noticeable that when the weight of PGD objective increases, the PGD accuracy improves but the clean accuracy decreases gradually. Hence, SOML with an increasing weight on the PGD objective could not dominate the MOML method.
>
> | Setting | Method | Clean Acc. | PGD Acc. |
> | :----: |:---- | :----: | :----: |
> | 5-shot *mini*-ImageNet | MAML+MOML | 55.66$\pm$0.78 | 39.38$\pm$0.77 |
> | 5-shot *mini*-ImageNet | MAML+SOML ($\gamma = 0.6$) | 50.54$\pm$0.72 | 37.37$\pm$0.71 |
> | 5-shot *mini*-ImageNet | MAML+SOML ($\gamma = 0.7$) | 47.98$\pm$0.73 | 38.55$\pm$0.73 |
> | 5-shot *mini*-ImageNet | MAML+SOML ($\gamma = 0.8$) | 47.90$\pm$0.68 | 38.24$\pm$0.66 |
> | 5-shot *mini*-ImageNet | MAML+SOML ($\gamma = 0.9$) | 47.03$\pm$0.70 | 39.02$\pm$0.68 |
> | 5-shot *mini*-ImageNet | MAML+SOML ($\gamma = 1.0$) | 46.83$\pm$0.71 | 39.63$\pm$0.71 |
> | 5-shot CUB | MAML+MOML | 67.57$\pm$0.78 | 55.26$\pm$0.87 |
> | 5-shot CUB | MAML+SOML ($\gamma = 0.6$) | 68.50$\pm$0.77 | 52.96$\pm$0.85 |
> | 5-shot CUB | MAML+SOML ($\gamma = 0.7$) | 67.80$\pm$0.70 | 53.16$\pm$0.84 |
> | 5-shot CUB | MAML+SOML ($\gamma = 0.8$) | 66.47$\pm$0.80 | 54.31$\pm$0.86 |
> | 5-shot CUB | MAML+SOML ($\gamma = 0.9$) | 65.06$\pm$0.84 | 55.14$\pm$0.87 |
> | 5-shot CUB | MAML+SOML ($\gamma = 1.0$) | 64.10$\pm$0.79 | 56.01$\pm$0.86 |
>
> **Q5**: Some details in the main text should be further clarified. For example, the definition of $\min g(z)$ is recommended to before or immediately after Eq. 1 and the role of $\alpha$ and $\omega$ of each cases should be briefly introduced.
>
> **A5**: We will further clarify those in detail in the revision.

---

### Official Review · Reviewer_7cN3 · 2021-07-13

**Rating:** 4
**Confidence:** 3

**Summary:**

This paper proposes a framework for multi objective bi-level optimization. They first prove (under assumptions) convergence of their iterative algorithm, then show several instances of problems in this framework.

**Limitations And Societal Impact:**

For limitations see main review.
No social impact discussed. Given the effort NeurIPS has put into this effort, I believe something should be added to discuss this.

**Main Review:**

I think the goal of multi task learning to meta-learning and has not been explored in depth as far as I am aware. The papers demonstrate this and introduce a general framework which I believe captures the problem.

I think discussion around why we might care about multi-task meta-learning and alternatives would be appreciated. In what settings, for example, do we have a distribution of tasks to meta-train over but not necessarily know which objectives we want at inner-test time? How does this relate to something more naive like joint training? In some sense, the extra level of meta- feels unneeded -- e.g. at meta-test time one could simply condition on the type objective with which to do well on and treat the problem like meta-learning not multi objective meta-learning. On the other hand, this is an interesting problem framing and thus hard to tell where clear benefits will be obtained. Show casing this further / building targeted examples where only multi-objective meta-learning will work would be interesting,

The problems discussed do not show the multi-objective nature of this work. My understanding is multi-objective method seek to find points on the pareto frontier of the different objectives. This is not actually shown or tested in the experiments. This is also a simpler setting than finding multiple points on this frontier which I traditionally associate with multi-objective and is what evolutionary methods often explore. Without visualizing this frontier though, it's hard to know if this method is working as intended or simply finding a different tradeoff.

The few shot learning setup: I found this a bit weird. Why the PGD attacks / adversarial robustness? I feel adversarial robustness adds a lot of complication to a problem when, as far as I can tell, it is simply another problem to test against. Have you considered leveraging something like this (https://arxiv.org/abs/1903.12261)?

If I am understanding things correctly, the vanilla baseline is trained targeting clean data where as SOML and MOML both have access to PGD data. It would be great to see a baseline for the vanilla baseline trained only with PGD data.

It is hard to evaluate performance / if this proposed solution is on the efficient frontier. It looks like adding MOL hurts clean performance by a fair bit, but does do better on PGD accuracy. A B-score (combination of PGD and clean data) is used, and is the main signal that the method proposed does better. If we are evaluating targeting a single objective, however, why not simply meta-train targeting this instead?

Also, how are the weights for SOML found? From the looks of it I don't see any search procedure done in the appendix. Again, if we know the target we are evaluating against I feel these could be set against that. Given enough compute, will SOML match MOML? or does the dynamic nature matter?


For the SSDA and multi task learning I have the same question on whether or not MOML is actually on the frontier.

Table 3 is also lacking error bars. All methods seem similar enough that these would be appreciated.

Theory:
This is treated as a big selling point and the result (thrm 2) is compelling but comes at the cost of some big assumptions which are not discussed. First, and most critically in my mind, K is never set to a large number. Usually it's < 10. In this setting, these problems can hardly be viewed as bi-level at all! Having some discussion on this would be appreciated to not mislead readers.

Clarity: I found the paper dense and put important details into the appendix.
In particular, the core of the method presented -- the MOPSolver -- is not discussed at all. I think this should be in the main text. Details of this algorithm, and it's connection to choosing weights ahead of time is important imo!
Additionally, details in how the weightings are chosen are also not discussed and should be in the main text.

There where also a number of awkward sentences e.g. (88 --"Multi-objective optimization is to address the problem of simultaneously minimizing multiple objectives").

Originality: As far as I know this method is novel though is closer to a novel combination of well-known techniques rather than an entirely novel method.


**Time Spent Reviewing:**

2

---

> ### Author Response · Authors · 2021-08-11
> **Response to Reviewer 7cN3**
>
> We thank you for the detailed and constructive comments. In the following we will make a response. If you think we have not addressed your concerns, we look forward to further discussions with you.
>
> **Q1**: It seems at meta-test time one could simply condition on the type objective with which to do well on and treat the problem like meta-learning not multi objective meta-learning.
>
> **A1**: We firstly emphasize that this paper studies meta learning from the perspective of bi-level optimization, which indicates it may not be simply understood from a task distribution perspective or it does not new tasks in the test process. For example, in the case of multi-task learning, the model parameterized by $\omega$ is directly using for test if the training process ends. Hence, it could not simply condition on the type objective in these cases and our proposed MOML considering the more general bi-level optimization problem can adapt to more practical application problems.
>
> **Q2**: Whether or not the proposed MOML can find a solution on the Pareto frontier?
>
> **A2**: To verify this, we take the 5-way 1-shot and 5-way 5-shot few-shot learning setting on the mini-ImageNet dataset as examples and we find the approximate Pareto frontier via grid search on the weights of MAML+SOML. The experimental results are shown in [https://www.dropbox.com/s/2fjbjx3ybz7ifzd/NeurIPS2021.pdf?dl=0](https://www.dropbox.com/s/2fjbjx3ybz7ifzd/NeurIPS2021.pdf?dl=0). According to the results, we can see that the solution of the proposed MOML falls very well on the Pareto frontier. Actually, there are similar results in the following reference [1]. We will add this discussion in the revision.
>
> [1] Multi-Task Learning as Multi-Objective Optimization. Sener et. al. NeurIPS, 2018.
>
> **Q3**: Why use the PGD attacks or adversarial robustness in the case of few-shot learning? Can you leverage some other types of robustness?
>
> **A3**: Firstly, as mentioned in our paper, it is natural to consider both the performance and the robustness in few-shot learning problems. Hence, we hope to learn an initialization which can be fast adopt to new tasks with an effective and robust model. Secondly, since the PGD attack is widely used in the robustness field, we add the robustness objective via the PGD attack into few-shot learning models, which can be formulated under the proposed MOML framework. Obviously, we can apply other methods to evaluate the robustness, such as the noise corruption robustness in the reference you provided, which will be studied in our future work.
>
> **Q4**: It is recommended to conduct a experiment of the vanilla baseline trained only with PGD data for the few-shot learning case.
>
> **A4**: In the following, we show the performance of the vanilla baseline (using MAML as an example) trained with PGD data only in its upper-level subproblem, where "MAML-PGD" means training MAML with a perturbed instead of a clean query set on the upper-level subproblem. According to the results, it is noticeable that the proposed MOML method performs superior to MOML-PGD.
>
> | Setting | Method | Clean Acc. | PGD Acc. |
> | :----: |:---- | :----: | :----: |
> | 1-shot *mini*-ImageNet | MAML-PGD | 25.24$\pm$0.52 | 22.95$\pm$0.51 |
> | 1-shot *mini*-ImageNet | MAML+MOML| 39.23$\pm$0.76 | 25.80$\pm$0.67 |
> | 5-shot *mini*-ImageNet | MAML-PGD | 46.83$\pm$0.72 | 39.63$\pm$0.68 |
> | 5-shot *mini*-ImageNet | MAML+MOML | 55.66$\pm$0.78 | 39.38$\pm$0.77 |
> | 1-shot CUB | MAML-PGD | 35.18$\pm$0.64 | 32.37$\pm$0.64 |
> | 1-shot CUB | MAML+MOML | 48.66$\pm$0.87 | 38.37$\pm$0.90 |
> | 5-shot CUB | MAML-PGD | 65.04$\pm$0.79 | 54.02$\pm$0.83 |
> | 5-shot CUB | MAML+MOML | 67.57$\pm$0.78 | 55.26$\pm$0.87 |
>
> **Q5**: Why not simply meta-train targeting B-score directly?
>
> **A5**: Firstly, B-score is just an auxiliary metric that we use to measure the trade-off between the clean and PGD accuracies in FSL and NAS problems. So we do not want to optimize the B-score. Secondly, directly optimizing B-score has no guarantee to obtain a Pareto optimal solution on the two objectives.
>
> **Q6**: How are the weights for SOML found?
>
> **A6**: In the case of semi-supervised domain adaptation, we directly apply the optimal weights used in the original method as the weights of SOML. In the other three cases (i.e., few-shot learning, multi-task learning, and neural architecture search), we first conduct a grid search for the weights of SOML and then select the best performance of SOML to report.
>
> **Q7**: Given enough compute, will SOML match MOML? or does the dynamic nature matter?
>
> **A7**: As shown in the answer to the second question (i.e., **A2**), MOML performs better than SOML with the best weight via the grid search. And there are similar results in the reference [1] mentioned in **A2**. Hence, it indicates MOML without manually tuning weights for each objective can achieve better performance than SOML with tuning weights and this can save a large computational cost.
>
> **Q8**: It is recommended to add the error bars in Table 3.
>
> **A8**: We will add the standard error in the revision.
>
> **Q9**: $K$ in Theorem 2 is not set to a large number.
>
> **A9**: In practice, the hypergradient is hard to calculate if $K$ is too large. Therefore, the bi-level optimization problem is usually solved with a small $K$ on the lower-level subproblem, such as in [2, 3, 4, 5].
>
> [2] Model-Agnostic Meta-Learning for Fast Adaptation of Deep Networks. Finn et. al. ICML, 2017.
>
> [3] Meta-Learning with Implicit Gradients. Rajeswaran et. al. NeurIPS, 2019.
>
> [4] DARTS: Differentiable Architecture Search. Liu et. al. ICLR, 2019.
>
> [5] Meta-Learning in Neural Networks: A Survey. Hospedales et. al.
>
> **Q10**: The details of MOPSolver and how the weightings are chosen are should be discussed in the main text.
>
> **A10**: We will add the details of MOPSolver and the choice of weights in the main text of the revision.

---

> > ### Comment · Reviewer_7cN3 · 2021-08-29
> > **Response**
> >
> > Thank you all for the detailed response + additional experiments. I think they will greatly strengthen the paper and do provide more convincing evidence. That being said, and after reading the other reviews, I plan to keep my score.

---

### Official Review · Reviewer_kG42 · 2021-07-15

**Rating:** 3
**Confidence:** 1

**Summary:**

The authors propose a Multi-Objection Meta Learning (MOML) formulation, offer a gradient based solver, show that the solver converges, and try to argue that MOML is useful in solving (1) few shot learning, (2) Semi-Supervised Domain Adaptatio, (3) Multi-Task Learning, and (4) NAS.

**Main Review:**

I find it very difficult to understand this paper, and I am not convinced of the contribution of this paper.

In equation 1, which is the main objective of this paper. What do the author mean by minimizing over a vector valued function? Are the authors trying to find the Pareto optimal solution? I wish that I could guess what the authors try to say by understanding their algorithm. However they again introduce a min over vector value function in equation (4). But the authors mention that "we can use any multi-objective optimization algorithms to solve it."

The simulations are not convincing me that MOML is better than the baselines. For example in section 5.1, the authors' method considered robustness against adversarial attack, but clearly MAML does not. It is not fair to compare a adversarial-attack-aware method with one that does not. Also the classification accuracy seems to have a big drop.




**Time Spent Reviewing:**

0.5

---

> ### Author Response · Authors · 2021-08-11
> **Response to Reviewer kG42**
>
> We thank you for the detailed and constructive comments. In the following we will make a response. If you think we have not addressed your concerns, we look forward to further discussions with you.
>
> **Q1**: What is the mean of minimizing over a vector valued function?
>
> **A1**: Minimizing over a vector-valued function is to find the solution in the Pareto-optimal solution set. We have explained this in Section 4.3 and Appendix A.1. We will further clarify those definitions in the revision.
>
> **Q2**: In the few-shot learning, it seems unfair comparison for MOML and baselines since baselines like MAML are not trained with adversarial data.
>
> **A2**: Since MAML trained with adversarial data usually has a lower clean accuracy, in the paper we only provide the results of the original MAML as a baseline. In the following table, we provide results of "MAML-PGD", which trains MAML with a perturbed instead of a clean query set on the upper-level subproblem. According to the results, we can see that MAML+MOML performs better than MAML-PGD. We will add these results to make a more comprehensive comparison in the revision.
>
> | Setting | Method | Clean Acc. | PGD Acc. |
> | :----: |:---- | :----: | :----: |
> | 1-shot *mini*-ImageNet | MAML-PGD | 25.24$\pm$0.52 | 22.95$\pm$0.51 |
> | 1-shot *mini*-ImageNet | MAML+MOML| 39.23$\pm$0.76 | 25.80$\pm$0.67 |
> | 5-shot *mini*-ImageNet | MAML-PGD | 46.83$\pm$0.72 | 39.63$\pm$0.68 |
> | 5-shot *mini*-ImageNet | MAML+MOML | 55.66$\pm$0.78 | 39.38$\pm$0.77 |
> | 1-shot CUB | MAML-PGD | 35.18$\pm$0.64 | 32.37$\pm$0.64 |
> | 1-shot CUB | MAML+MOML | 48.66$\pm$0.87 | 38.37$\pm$0.90 |
> | 5-shot CUB | MAML-PGD | 65.04$\pm$0.79 | 54.02$\pm$0.83 |
> | 5-shot CUB | MAML+MOML | 67.57$\pm$0.78 | 55.26$\pm$0.87 |

---

> > ### Comment · Reviewer_kG42 · 2021-09-03
> > **Response**
> >
> > Thank the authors for the clarifications. While the authors do address some of my clarity concerns, I tend to keep my current score. For example, I still do not understand why we have to mix the step-up of few-shot learning and adversarial robustness; the authors clarify that the objective is to find Pareto-optimal solution set now, I wish the authors could directly discuss the benefit of multi-objective optimization (i.e., motivation of doing multi-objective optimization); etc. Overall I think the clarify of this paper can be drastically improved so that the readers can appreciate the contributions more.

---

### Decision · Program_Chairs · 2021-09-28

**Decision:**

Accept (Poster)

**Comment:**

Unfortunately, all reviewers believe that the paper is not ready for publication at this point. The response provided by the authors partially addressed some of the comments raised by the reviewers, but it did not convince them to raise their score. I encourage the authors to revise their paper based on the comments provided by the reviewers and consider submitting their work to upcoming venues.

**Consistency Experiment:**

NeurIPS has a long history of experimentation. In 2014, NeurIPS ran an experiment in which 10% of submissions were reviewed by two independent committees to quantify the randomness in the review process. This year, we repeated a variant of this experiment to see how the quality of the review process has changed over time.  This paper was part of the experiment and was therefore assigned to two committees (consisting of reviewers, an Area Chair, and a Senior Area Chair) that reached independent decisions.  If both committees made the same recommendation, this recommendation was followed. If a single committee recommended acceptance, the paper was accepted (with the exception of a few cases in which the other committee identified what we considered a fatal flaw, e.g., an error in a key result).

This copy’s committee reached the following decision: **Reject**

The other committee assigned to the paper recommended **Accept (Poster)**.  You can find the other set of reviews, along with any follow up discussion with the authors here:
https://openreview.net/forum?id=wKf9iSu_TEm